# Sparse Model Soups: A Recipe for Improved Pruning via Model Averaging

**Max Zimmer[1], Christoph Spiegel[1] & Sebastian Pokutta[1,2]**
[1]Department for AI in Society, Science, and Technology, Zuse Institute Berlin, Germany
[2]Institute of Mathematics, Technische Universität Berlin, Germany
`{zimmer,spiegel,pokutta}@zib.de`

## Abstract

Neural networks can be significantly compressed by *pruning*, yielding *sparse* models with reduced storage and computational demands while preserving predictive performance. *Model soups* (Wortsman et al., 2022a) enhance generalization and out-of-distribution (OOD) performance by averaging the parameters of multiple models into a single one, without increasing inference time. However, achieving both sparsity and parameter averaging is challenging as averaging arbitrary sparse models reduces the overall sparsity due to differing sparse connectivities. This work addresses these challenges by demonstrating that exploring a single retraining phase of *Iterative Magnitude Pruning* (IMP) with varied hyperparameter configurations such as batch ordering or weight decay yields models suitable for averaging, sharing identical sparse connectivity by design. Averaging these models significantly enhances generalization and OOD performance over their individual counterparts. Building on this, we introduce Sparse Model Soups (SMS), a novel method for merging sparse models by initiating each prune-retrain cycle with the averaged model from the previous phase. SMS preserves sparsity, exploits sparse network benefits, is modular and fully parallelizable, and substantially improves IMP's performance. We further demonstrate that SMS can be adapted to enhance state-of-the-art pruning-during-training approaches.

## 1 Introduction

State-of-the-art Neural Network architectures typically rely on extensive over-parameterization with millions or billions of parameters (Zhang et al., 2016). In consequence, these models have significant memory requirements and the training and inference process is computationally demanding. However, recent work (e.g. Han et al., 2015; Lin et al., 2020; Renda et al., 2020; Zimmer et al., 2022) has demonstrated that these resource demands can be significantly reduced by *pruning* the model, i.e., removing redundant structures such as individual parameters or groups thereof. The resulting *sparse* models demand considerably less storage and floating-point operations (FLOPs) during inference, while retaining performance comparable to *dense* models.

A different line of research has shown that the performance of a predictor can be significantly enhanced by leveraging multiple models, instead of selecting the best one on a hold-out validation dataset and discarding the rest. Such *ensembles* combine the predictions of $m \in \mathbb{N}$ individually trained models by averaging their output predictions (Ganaie et al., 2021; Mehrtash et al., 2020; Chandak et al., 2023; Fort et al., 2019). Prediction ensembles have been shown to improve the predictive performance and positively impact predictive uncertainty metrics such as calibration, out-of-distribution generalization as well as model fairness (Lakshminarayanan et al., 2017; Mehrtash et al., 2020; Allen-Zhu & Li, 2023; Ko et al., 2023). A significant drawback of ensembling is that all models have to be evaluated during deployment: the inference costs are hence increasing by a factor of $m$, a problem that has partially been addressed by leveraging an ensemble of sparsified, more efficient models (Liu et al., 2021; Whitaker & Whitley, 2022; Kobayashi et al., 2022).

Several works propose to instead average the parameters (Izmailov et al., 2018; Wortsman et al., 2022a; Rame et al., 2022; Matena & Raffel, 2022), constructing a single model for inference. Unlike prediction ensembles that require sufficiently diverse models for better performance, such

*Model Soups* (Wortsman et al., 2022a) need models to lie in a linearly connected basin of the loss landscape. However, training models from scratch with differing random seeds but identical initialization often yields models whose parameter average will perform much worse than the individual models (Neyshabur et al., 2020) with recent studies investigating neuron permutation to align them within a single basin (Singh & Jaggi, 2020; Ainsworth et al., 2023). Beyond the initial challenge of identifying networks suitable for averaging, another problem emerges when attempting to leverage the computational advantages of sparse networks: averaging models with different sparse connectivities reduces overall sparsity (cf. Figure 1) and may require to prune again (Yin et al., 2022a;b), potentially resulting in further performance degradation.

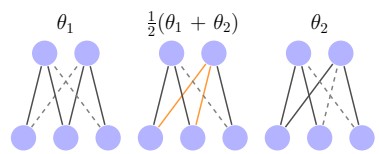

In this work, we tackle the challenge of concurrently leveraging sparsity as well as the benefits of combining multiple models into a single one. We draw our inspiration from recent work in the domain of *transfer learning* (Neyshabur et al., 2020; Wortsman et al., 2022a; Rame et al., 2022), which has shown that fine-tuning multiple copies of a pretrained model, differing only in random seed, yields models sufficiently similar for averaging and sufficiently diverse for generalization improvements. At the core of our work lies the observation that a single prune-retrain phase in standard *prune after training* strategies, such as ITERATIVE MAGNITUDE PRUNING (IMP, Han et al., 2015), closely resembles the transfer learning paradigm. Starting from a pretrained model, the optimization objective shifts abruptly, either due to a new target domain or

Figure 1: Creating the average (middle) of two networks with different sparsity patterns (left, right) may lower overall sparsity, changing pruned weights (dashed) to non-zero (solid), with reactivated weights highlighted in orange.

subspace constraints imposed by pruning, followed by a training process termed 'fine-tuning', often used interchangeably with 'retraining' to recover from pruning (Hoefler et al., 2021).

We find that, akin to the fine-tuning phase in transfer learning, exploring various hyperparameter configurations during the retraining phase after pruning generates models that are suitable for averaging while sharing the same sparse connectivity by design. Such sparse averages exhibit superior performance compared to both their individual counterparts as well as to models retrained $m$ times as long, effectively reducing IMP's runtime. Additionally, we initiate the next prune-retrain cycle from the averaged model just obtained, which remarkably also enhances the performance of the individual retraining runs before averaging again. Our proposed approach, SPARSE MODEL SOUPS (SMS), tackles the aforementioned challenges and enables inference complexity independent of $m$, utilizes pretrained models without requiring training from scratch, preserves the sparsity pattern while leveraging sparsity benefits, and considerably improves IMP's generalization and OOD performance.

**Contributions.** To summarize, our contributions can be stated as follows.

1. We demonstrate that pruning a well-trained model and retraining multiple copies with varied hyperparameter like batch ordering, weight decay, or retraining duration and length, produces models suitable for constructing an averaged model which exhibits superior generalization and OOD performance compared to its individual components. Importantly, these models retain the sparsity pattern of their pruned parent, preserved in their parameter average.

2. We propose *Sparse Model Soups* (SMS), a novel method for merging sparse models into a single classifier, leveraging the idea of starting each prune-retrain phase of IMP from an averaged model. SMS significantly enhances the performance of IMP in two ways: first, the average improves upon the individual models in terms of generalization and OOD performance and, secondly, the models retrained from an average exhibit better performance compared to those retrained from a single model.

3. We extend our findings to the *pruning during training* domain, demonstrating SMS's versatility by integrating it with multiple other state-of-the-art approaches. This yields substantial performance improvements and enhances their competitiveness in comparison to other leading methods that sparsify during training.

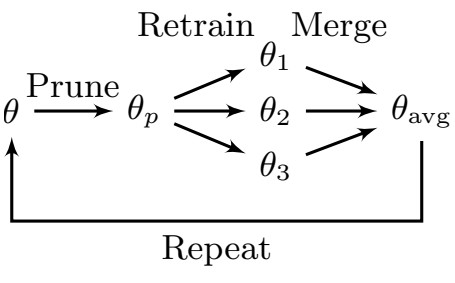

**Algorithm 1** Sparse Model Soups

**Input:** Pretrained model $\theta$
**Output:** Sparse model soup
 1: **for** each prune-retrain cycle **do**
 2:     Prune $\theta$
 3:     **for** $i \leftarrow 1$ to $m$ **do**          ▷ Fully parallelizable
 4:         $\theta_i \leftarrow \theta$
 5:         Retrain $\theta_i$ with specific hyperparameters
 6:     **end for**
 7:     $\theta \leftarrow \text{Merge}(\theta_1, \ldots, \theta_m)$
 8: **end for**
 9: **return** $\theta$

Figure 2: Left: Sketch of the algorithm for a single phase and $m = 3$. Right: Pseudocode for SMS. Merge($\cdot$) takes $m$ models as input and returns a linear combination of the models (cf. Section 2.2).

**Outline.** We introduce our framework in Section 2. In Section 3, we experimentally validate our findings across image classification, semantic segmentation, and neural machine translation architectures and datasets. Section 4 reviews relevant literature, followed by a discussion in Section 5.

## 2 METHODOLOGY: SPARSE MODEL SOUPS

### 2.1 PRELIMINARIES

Our focus lies on *model pruning* which aims at removing individual weights as exemplified by the previously introduced IMP approach. IMP, a *prune after training* algorithm, follows a three-stage pipeline. It starts with a pretrained model parameterized by $\theta$, prunes weights with magnitudes below a certain threshold, and then restores predictive power through retraining. This prune-retrain cycle is repeated multiple times, with each pruning step's threshold determined by the suitable percentile to achieve the desired target sparsity after a predefined number of such phases. Recent studies (Gale et al., 2019; Zimmer et al., 2023) have demonstrated that magnitude pruning results in sparse models with performance competitive to more complex algorithms.

Given $m$ sparse models $f_{\theta_i}$ with weights $\theta_i \in \mathbb{R}^n, i \in \{1, \ldots, m\}$, prediction ensembles construct a model as the functional equivalent of the average of the models' output (Liu et al., 2021; Whitaker & Whitley, 2022; Kobayashi et al., 2022). This ensemble requires $m$ forward passes for evaluation, but maintains the overall sparsity level. In contrast, our focus lies on examining the performance and sparsity of a single model, specifically a linear combination of other models. Given scalars $\lambda_i \in \mathbb{R}$, we consider the prediction function $f_{\bar{\theta}}$, parameterized by the weights given by

$$\bar{\theta} = \sum_{1 \leq i \leq m} \lambda_i \theta_i. \tag{1}$$

A special case occurs when $\lambda_i = 1/m$ for all $i$, resulting in $\bar{\theta}$ representing the average of all networks. Averaging the weights of arbitrary sparse models can result in reduced overall sparsity, as different networks may possess distinct sparse connectivities, causing the averaging process to eliminate zeros from the tensors (cf. Figure 1). Yin et al. (2022b) and Yin et al. (2022a) address this issue by pruning $\bar{\theta}$ to align with the original networks' sparsity levels. However, this approach has a notable drawback: if the sparsity patterns differ significantly, pruning-induced performance degradation may occur.

### 2.2 SPARSE MODEL SOUPS

Inspired by recent advancements in the transfer learning domain (Neyshabur et al., 2020; Wortsman et al., 2022a), which demonstrate that models fine-tuned from the same pretrained model end up in the same loss basin and can be combined into a *soup*, we hypothesize that a similar behavior can be achieved during retraining from the same pruned model. Our motivation stems from the resemblance between the transfer learning paradigm and a single phase of IMP. When transitioning from the

source to the target domain, the optimization objective changes abruptly, requiring adaptation (i.e., fine-tuning) to minimize the new objective. Similarly, 'hard' pruning alters the loss abruptly and requires adapting (i.e., retraining) given the newly added sparsity constraints.

A single phase of this idea is illustrated on the left of Figure 2. The pretrained model's weights $\theta$ are pruned, yielding model $\theta_p$, which is then replicated $m$ times. In this setup, pruned weights remain permanently non-trainable. Subsequently, each of the $m$ models is independently retrained with different hyperparameter configurations, such as varying random seeds, weight decay factors, retraining lengths, or learning rate schedules. Finally, the $m$ retrained models are merged into a single model. This process ensures that all $m$ retrained models $\theta_1, \ldots, \theta_m$ share the same sparsity pattern, as they all originate from the same pruned network with a fixed pruning mask. However, when combining models after multiple prune-retrain cycles, identical sparsity connectivity between all models is not guaranteed. To address this, we average the models after each phase and begin the subsequent phase with the previously averaged model. The resulting method, termed SPARSE MODEL SOUPS (SMS), is presented as pseudocode on the right of Figure 2.

SMS offers several benefits and addresses key challenges. First, the inference complexity of the final model remains independent of $m$. The method is highly modular, allowing for different hyperparameter configurations and different $m$ in each phase. Further, the retraining of the $m$ models can be fully parallelized, enhancing efficiency as detailed in Section 3.2. By initiating each phase with the merged model from the previous one, sparsity patterns are preserved, and the advantages of sparse networks are utilized; as the number of cycles increases, the networks become sparser, potentially leading to further efficiency gains. Moreover, SMS effectively leverages the benefits of large pretrained models without the need for training from scratch.

Effectively merging models for enhanced generalization can be challenging, as models may end up far apart. We primarily employ two convex combination methods from Wortsman et al. (2022a): *UniformSoup* and *GreedySoup*. UniformSoup equally weighs each model with $\lambda_i = 1/m$. On the other hand, GreedySoup orders models by validation accuracy, sequentially including models only if they improve validation accuracy over the prior subset.

## 3 EXPERIMENTAL RESULTS

We first outline our general experimental approach. For reproducibility, our implementation is available at github.com/ZIB-IOL/SMS. We evaluate our approach on well-known datasets for image recognition, semantic segmentation, and neural machine translation (NMT), including *ImageNet-1K* (Russakovsky et al., 2015), *CIFAR-10/100* (Krizhevsky et al., 2009), *Celeb-A* (Liu et al., 2015), *CityScapes* (Cordts et al., 2016), *WMT16 DE-EN* (Bojar et al., 2016) and the benchmark OOD-datasets *CIFAR-100-C* and *ImageNet-C* (Hendrycks & Dietterich, 2019). We utilize state-of-the-art architectures, such as *ResNets* (He et al., 2015), *WideResNets* (Zagoruyko & Komodakis, 2016), *MaxViT* (Tu et al., 2022), *PSPNet* (Zhao et al., 2017), and the *T5* transformer (Raffel et al., 2020). For validation, we use 10% of the training data. We use magnitude-based unstructured pruning and filter norm-based structured pruning as suggested by Li et al. (2016). For retraining, we stick to the linear learning rate schedules LLR and ALLR (Zimmer et al., 2023), with further details in Appendix A.2. Appendix A describes exact hyperparameters and settings for pretraining, pruning and retraining.

Recomputing *Batch-Normalization* (BN) (Ioffe & Szegedy, 2015) statistics is crucial in both pruning and model averaging, as observed by Li et al. (2020) and Jordan et al. (2022), respectively. When reporting test accuracy for single or averaged models, we reset all BN layers and recompute statistics using a forward pass on the entire training dataset.

### 3.1 EVALUATING SPARSE MODEL SOUPS

We evaluate SMS against key baselines, beginning with a comparison at each prune-retrain phase to the best-performing single model among all averaging candidates (*best candidate*), the mean accuracy of these candidates (*mean candidate*) and *regular IMP* (i.e., $m = 1$). From the second phase onwards, averaging candidates are retrained from a previous model soup, distinguishing *best candidate* from regular IMP without averaging. Given the lower computational demands of regular IMP compared to SMS, which trains $m$ models per phase and hence increases the total number of retraining epochs by a factor of $m$, we also contrast SMS with an extended IMP version retrained $m$ times as long (*IMP$_{m\times}$*).

Table 1: WideResNet-20 on CIFAR-100 and ResNet-50 on ImageNet: Test accuracy comparison for target sparsities 98% (top) and 90% (bottom) given three prune-retrain cycles. We report results using UniformSoup as well as GreedySoup for merging. Results are averaged over multiple seeds with standard deviation included. The best value is highlighted in bold.

**CIFAR-100** (98%)

| Accuracy of | Sparsity 72.8% (Phase 1) | | | Sparsity 92.6% (Phase 2) | | | Sparsity 98.0% (Phase 3) | | |
|---|---|---|---|---|---|---|---|---|---|
| | $m=3$ | $m=5$ | $m=10$ | $m=3$ | $m=5$ | $m=10$ | $m=3$ | $m=5$ | $m=10$ |
| **SMS** (uniform) | **76.50 ±0.16** | **76.59 ±0.13** | **76.75 ±0.28** | **75.55 ±0.60** | **76.19 ±0.37** | **76.21 ±0.43** | **72.67 ±0.29** | **72.90 ±0.64** | **73.05 ±0.45** |
| best candidate | 75.58 ±0.19 | 75.71 ±0.08 | 75.96 ±0.13 | 74.51 ±0.47 | 75.01 ±0.74 | 75.00 ±0.34 | 71.77 ±0.04 | 71.77 ±0.37 | 72.21 ±0.02 |
| mean candidate | 75.37 ±0.12 | 75.58 ±0.03 | 75.55 ±0.26 | 74.32 ±0.40 | 74.71 ±0.48 | 74.70 ±0.42 | 71.41 ±0.09 | 71.61 ±0.40 | 71.66 ±0.19 |
| **SMS** (greedy) | 76.06 ±0.69 | 76.43 ±0.24 | 76.60 ±0.47 | 75.34 ±0.15 | 75.39 ±0.44 | 75.51 ±0.66 | 72.08 ±0.23 | 71.86 ±0.64 | 72.44 ±0.20 |
| best candidate | 75.58 ±0.19 | 75.65 ±0.00 | 75.94 ±0.15 | 74.85 ±0.04 | 74.53 ±0.42 | 74.57 ±0.21 | 71.05 ±0.43 | 71.01 ±0.49 | 71.47 ±0.23 |
| mean candidate | 75.37 ±0.12 | 75.54 ±0.03 | 75.54 ±0.27 | 74.52 ±0.25 | 74.27 ±0.52 | 74.20 ±0.31 | 70.84 ±0.41 | 70.69 ±0.75 | 70.87 ±0.01 |
| IMP$_{m\times}$ | 75.85 ±0.26 | 76.05 ±0.00 | 75.76 ±0.24 | 74.09 ±0.24 | 74.19 ±0.44 | 74.74 ±0.06 | 70.92 ±0.07 | 70.31 ±0.52 | 71.85 ±0.15 |
| IMP-RePrune | | — N/A — | | | — N/A — | | 68.19 ±0.44 | 65.53 ±0.06 | 63.62 ±0.90 |
| IMP | | — 75.54 ±0.41 — | | | — 74.09 ±0.13 — | | | — 70.74 ±0.08 — | |

**ImageNet** (90%)

| Accuracy of | Sparsity 53.6% (Phase 1) | | | Sparsity 78.5% (Phase 2) | | | Sparsity 90.0% (Phase 3) | | |
|---|---|---|---|---|---|---|---|---|---|
| | $m=3$ | $m=5$ | $m=10$ | $m=3$ | $m=5$ | $m=10$ | $m=3$ | $m=5$ | $m=10$ |
| **SMS** (uniform) | **76.74 ±0.20** | 76.89 ±0.18 | **77.01 ±0.05** | 76.04 ±0.21 | 76.30 ±0.13 | **76.49 ±0.12** | 74.53 ±0.04 | **74.82 ±0.08** | **74.96 ±0.16** |
| best candidate | 76.07 ±0.01 | 76.07 ±0.21 | 76.14 ±0.18 | 75.48 ±0.16 | 75.46 ±0.11 | 75.70 ±0.03 | 74.00 ±0.03 | 74.19 ±0.08 | 74.25 ±0.13 |
| mean candidate | 75.99 ±0.04 | 75.95 ±0.14 | 75.96 ±0.08 | 75.40 ±0.11 | 75.42 ±0.10 | 75.55 ±0.05 | 73.94 ±0.03 | 74.11 ±0.11 | 74.13 ±0.12 |
| **SMS** (greedy) | 76.74 ±0.19 | **76.92 ±0.15** | 76.88 ±0.11 | **76.12 ±0.18** | **76.35 ±0.21** | 76.11 ±0.26 | **74.58 ±0.03** | 74.77 ±0.03 | 74.52 ±0.11 |
| best candidate | 76.08 ±0.01 | 76.08 ±0.21 | 76.14 ±0.18 | 75.48 ±0.18 | 75.53 ±0.24 | 75.34 ±0.19 | 74.03 ±0.11 | 74.21 ±0.00 | 73.95 ±0.07 |
| mean candidate | 75.98 ±0.04 | 75.95 ±0.14 | 75.95 ±0.08 | 75.42 ±0.15 | 75.45 ±0.21 | 75.24 ±0.17 | 73.94 ±0.01 | 74.09 ±0.03 | 73.76 ±0.12 |
| IMP$_{m\times}$ | 76.25 ±0.08 | 76.21 ±0.14 | 76.46 ±0.04 | 75.74 ±0.03 | 75.87 ±0.11 | 75.93 ±0.03 | 74.34 ±0.09 | 74.56 ±0.24 | 74.50 ±0.09 |
| IMP-RePrune | | — N/A — | | | — N/A — | | 72.97 ±0.25 | 72.58 ±0.01 | 72.08 ±0.12 |
| IMP | | — 75.97 ±0.16 — | | | — 75.19 ±0.14 — | | | — 73.59 ±0.04 — | |

Unlike SMS, IMP$_{m\times}$ cannot be parallelized in each phase as the extended number of epochs are executed sequentially, yet, the overall computational and memory requirements remain identical between both methods. We also compare SMS to another baseline termed *IMP-RePrune*, where regular IMP is executed $m$ times and model averaging is performed after the final phase. Unlike SMS, which merges after every phase and hence maintains a consistent pruning mask, the individual models in IMP-RePrune may develop diverging pruning masks over multiple phases, potentially reducing the overall sparsity when averaged. To ensure comparable sparsity levels, IMP-RePrune incorporates a repruning step to address any sparsity reduction after averaging (Yin et al., 2022b).

Table 1 presents results for three-phase IMP using WideResNet-20 on CIFAR-100 and ResNet-50 on ImageNet, employing ALLR and ten retraining epochs per phase. For SMS, we vary the random seeds across each of the $m$ models. The three main columns correspond to phases and sparsities, targeting 98% sparsity for CIFAR-100 and 90% for ImageNet. Each main column has three subcolumns, indicating the number of models to average (3, 5, or 10). We discuss the main observations below.

1. **SMS significantly enhances generalization.** We find that SMS consistently improves upon the test accuracy of the best candidate, often with a 1% or higher margin. This confirms that the models after retraining are averageable, resulting in better generalization than individual models. SMS notably improves upon both regular IMP and its extended retraining variant, IMP$_{m\times}$, with up to 2% enhancements even when using $m=3$ splits.

2. **Starting from a model soup enhances generalization.** Surprisingly, the best candidates in the second and third phase frequently exceed both IMP and IMP$_{m\times}$. While some improvement is anticipated when picking the best among multiple candidates, it is notable that the mean candidate accuracy (i.e., mean candidate) often surpasses both IMP and IMP$_{m\times}$ as well. This suggests that initiating from a soup, as opposed to starting from a singular model as in regular IMP, enhances generalization in the subsequent phase.

3. **IMP-RePrune faces sparsity reduction and performance degradation.** Naively averaging IMP's models in the final phase often leads to reduced sparsity due to differing sparse connectivities, requiring repruning which typically degrades performance compared to individual models. Under certain conditions, IMP-RePrune can remain competitive, suggesting that similar pruning patterns may emerge across multiple pruning rounds (cf. Appendix B.1).

In summary, we find that averaging after each phase and starting subsequent phases from the soup of the previous phase capitalizes on two dynamics, which often enable significant improvements over IMP: first of all, the model soup consistently improves upon individual soup candidates, demonstrating that pruned and retrained models are indeed averageable and exhibit enhanced generalization. Secondly, models retrained from a pruned soup also outperform those following the classical prune-retrain cycle. Pruning a model with higher generalization performance yields better models after retraining, despite experiencing a larger pruning-induced performance drop. For full results on different architectures, datasets, target sparsities, and structured pruning, we refer to Appendix B.1.

Table 1 also contrasts uniform and greedy soup selections. With just the random seed varied for the $m$ models, none appear to diverge to a different basin, rendering greedy subset selection unnecessary. The uniform approach predominantly outperforms the greedy one, notably when comparing the best or mean candidates in scenarios like the last phase of CIFAR-100, indicating that retraining from previous greedy soups yields less performant models.

## 3.2 EXAMINING SPARSE MODEL MERGING

Having established the merits of SMS, we now investigate its success and limitations in more detail.

**Exploring parameters beyond random seeds.** Previously, we focused on varying the random seed for simplicity. Figure 3 presents a scatter plot for One Shot IMP (70% sparsity) of ResNet-50 trained on ImageNet, comparing the effects of varying the random seed, weight decay strength, retraining duration, and initial learning rate of a linearly decaying schedule. Exact hyperparameters are listed in Appendix B.2. Parameter averages are constructed from two-element pairs of models in the uniform soup setting. The plot displays the test accuracy of the averaged model versus the maximal test accuracy of each pair. In this setting, most averaged models show a net improvement over their individual components, demonstrating that varying different hyperparameters in the retraining phase of One Shot IMP produces models within the same loss basin. Comparing different hyperparameters, the most substantial and consistent improvement comes from varying the random seed. Unlike the random seed, which only introduces variability due to inherent randomness, other parameters such as weight decay have a direct, controllable impact on the results; for instance, a poorly chosen weight decay value could significantly degrade performance.

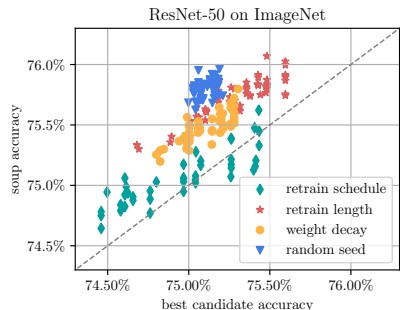

Figure 3: Accuracy of average of two models vs. the maximal individual accuracy. All models are pruned to 70% sparsity (One Shot) and retrained, varying the indicated hyperparameters.

**OOD-robustness and fairness.** We also explored if SMS enhances robustness to out-of-distribution data, akin to regular model soups (Wortsman et al., 2022a), using benchmark robustness datasets CIFAR-100-C and ImageNet-C (Hendrycks & Dietterich, 2019) for evaluation. SMS consistently outperformed individual models from IMP and other baselines, displaying better resilience to common corruptions, especially with ImageNet-C, which showed up to a 2.5% increase in OOD-accuracy (see Figure 8, Table 14 in Appendix B.2.3 for more details). Further, previous research suggests that pruning can exacerbate unfairness across data subgroups (Hooker et al., 2019; 2020; Paganini, 2020). In Appendix B.2.4, we examine if SMS could alleviate such pruning-induced unfairness, and found SMS to exhibit less severe negative impacts on individual subgroups than IMP.

**Instability to randomness and recovering it.** Neyshabur et al. (2020) demonstrated that during training from scratch, the inherent randomness in batch selection alone suffices to cause divergence between two models to the extent that they are not averageable, even when starting from identical (random) initialization. Such *instability to randomness* can be mitigated by ensuring sufficient pretraining: Frankle et al. (2020) specifically analyze the amount of training required before splitting a network into two copies further trained with different random seeds, such that the final models reside within a linearly connected basin. In that vein, several works (Frankle et al., 2020; Evci et al., 2022; Paul et al., 2023) study the stability of IMP with weight rewinding (IMP-WR) in the context of the Lottery Ticket Hypothesis (Frankle & Carbin, 2018). In contrast, we explore retraining networks

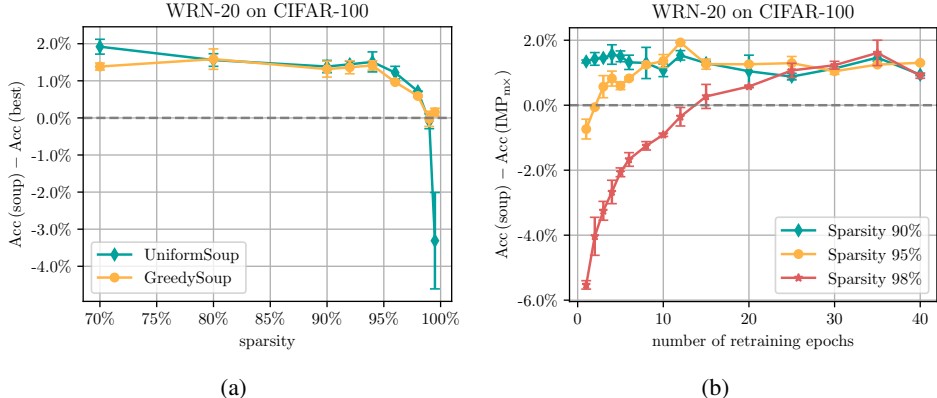

Figure 4: WideResNet-20 on CIFAR-100: (a) Accuracy difference between the soup ($m = 5$) and best averaging candidate after One Shot pruning and retraining for varying sparsity levels. (b) Accuracy difference between the soup ($m = 3$) and IMP$_{3\times}$ retrained three times as long as indicated on the x-axis, using One Shot pruning to 90%, 95% and 98% sparsity. Results are averaged over multiple random seeds with min-max bands indicated.

without rewinding. Based on the aforementioned instability analysis, we conjecture that different splits of a pruned network converge to a common basin under low sparsity and moderate learning rates, while high pruning levels may potentially reduce stability to randomness.

Figure 4a shows the difference in test accuracy between a soup of $m = 5$ models and the best candidate at different sparsity levels. Each point corresponds to the best configuration across varying retraining lengths (5, 20, and 50 epochs) and schedules (LLR and ALLR). UniformSoup and GreedySoup enhance accuracy by up to 2% over individual models. Yet, as sparsity increases, this benefit declines, with UniformSoup collapsing in performance. Beyond a certain sparsity, stability to randomness declines, causing model divergence and hindering beneficial model averaging. Unlike UniformSoup, GreedySoup performs at least as well as the best individual model.

Similarly, Figure 4b depicts the difference between the model soup ($m = 3$) and IMP$_{3\times}$, where the latter is retrained for $m \cdot k$ epochs and the former trains $m$ models for $k$ epochs, with $k$ denoted on the x-axis. Again, we plot the best configuration varying the retraining schedule (LLR, ALLR) and merging method (UniformSoup, GreedySoup). For moderate sparsity (green), averaging $m$ models is more effective than training a single model $m$-times as long, even with brief retraining. At high sparsity (red), short-term retraining and averaging $m$ models underperforms compared to extending a single model's training. However, a break-even point emerges around 15 epochs, beyond which the benefit of extended single model training diminishes, with the $m$ models sufficiently trained for merging. In iterative pruning, we expect SMS to require fewer retraining epochs per phase, benefiting from the gradual sparsity increment. Despite extensive retraining, IMP$_{m\times}$ fails to match SMS.

**Efficiency of SMS.** Each IMP-cycle consists of $k$ epochs, while IMP$_{m\times}$ extends this to $m \cdot k$ epochs sequentially. Contrarily, SMS executes each phase with $m$ distinct models, independently trained for $k$ epochs, allowing parallelization that can lower wall-time by a factor of $1/m$ compared to IMP$_{m\times}$. Nevertheless, the overall compute and memory requirements for both IMP$_{m\times}$ and SMS remain at $m \cdot k$ units. Resource-wise, IMP$_{m\times}$ and SMS are hence on par; however, the parallelization in SMS underscores a practical advantage. Further, our data shows SMS often significantly outperforming IMP$_{m\times}$, suggesting comparable accuracy can be achieved with fewer total retraining epochs.

In Appendix C, we conduct ablation studies to individually assess the impact of the retraining schedule and the number of epochs per phase during the execution of SMS.

### 3.3 Improving Pruning during Training algorithms

We extend our findings to magnitude-pruning based methods within the *pruning during training* domain. These methods, unlike IMP, start with a randomly initialized model and sparsify it during regular training. We focus on GMP (Zhu & Gupta, 2017; Gale et al., 2019), DPF (Lin et al., 2020),

Table 2: ResNet-50 on ImageNet: Comparison of BIMP, GMP and DPF with their SMS-extended variants for goal sparsity levels of 70%, 80% and 90%. Each subcolumn denotes the top-1 accuracy and the theoretical speedup at a given sparsity. All results are averaged over multiple seeds and include standard deviations. The **best**, second best, and third best values are highlighted.

**ImageNet**

| Method | Sparsity 70% | | Sparsity 80% | | Sparsity 90% | |
|---|---|---|---|---|---|---|
| | Accuracy | Speedup | Accuracy | Speedup | Accuracy | Speedup |
| **BIMP+SMS** | 76.20 ±0.09 | 2.7 ±0.0 | 75.76 ±0.11 | 3.7 ±0.0 | 74.05 ±0.02 | 6.1 ±0.0 |
| BIMP | 75.62 ±0.02 | 2.7 ±0.0 | 75.08 ±0.16 | 3.7 ±0.0 | 73.53 ±0.05 | 6.1 ±0.0 |
| **GMP+SMS** | 75.10 ±0.00 | 2.7 ±0.0 | 74.48 ±0.00 | 3.9 ±0.0 | 73.12 ±0.02 | 7.7 ±0.0 |
| GMP | 74.55 ±0.07 | 2.7 ±0.0 | 73.92 ±0.12 | 4.0 ±0.0 | 72.81 ±0.00 | 7.0 ±0.0 |
| **DPF+SMS** | 76.26 ±0.10 | 2.7 ±0.0 | 75.85 ±0.05 | 3.6 ±0.0 | 74.31 ±0.00 | 6.0 ±0.0 |
| DPF | 75.74 ±0.02 | 2.6 ±0.0 | 75.27 ±0.02 | 3.6 ±0.0 | 73.88 ±0.01 | 5.9 ±0.0 |
| GSM | 73.69 ±0.70 | 2.9 ±0.1 | 72.75 ±0.62 | 4.5 ±0.3 | 70.08 ±0.94 | 9.5 ±0.8 |
| DNW | 75.81 ±0.05 | 2.5 ±0.0 | 75.35 ±0.21 | 3.3 ±0.0 | 74.24 ±0.12 | 5.5 ±0.1 |
| LC | 75.03 ±0.20 | 2.4 ±0.0 | 73.87 ±0.62 | 3.2 ±0.0 | 67.57 ±2.71 | 5.1 ±0.0 |
| DST | 72.47 ±0.01 | 4.1 ±0.0 | 72.32 ±0.03 | 9.7 ±0.3 | 71.35 ±0.09 | 13.2 ±0.4 |

and BIMP (Zimmer et al., 2023). GMP applies a pruning schedule to iteratively update a pruning mask throughout training. DPF, while following the same schedule, enables error compensation by updating the dense parameters using the pruned model's gradient. BIMP divides the training budget into a pretraining phase and multiple IMP cycles thereafter, rendering it closest to IMP.

These three approaches can be easily adapted using SMS. BIMP can integrate SMS within individual phases. Both GMP and DPF prune at uniformly distributed timesteps during training. We regard the interval between two such steps as a phase, during which we create $m$ copies of the recently pruned model, train them with different random seeds, and merge them before the next pruning step. Unlike IMP or BIMP, GMP and DPF follow the original learning rate schedule throughout a phase.

Table 2 compares accuracy and sparsity-induced theoretical speedup (Blalock et al., 2020) of the three methods and their SMS-enhanced versions to other state-of-the-art pruning during training methods like GSM (Ding et al., 2019), DNW (Wortsman et al., 2019), LC (Carreira-Perpiñán & Idelbayev, 2018), and DST (Liu et al., 2020). For a fair comparison, we solely consider the *dense-to-sparse* training paradigm, as opposed to *pruning at initialization* (Lee et al., 2019; Tanaka et al., 2020) or *dynamic sparse training* (DST, Mocanu et al., 2018; Dettmers & Zettlemoyer, 2019; Evci et al., 2020) methods. We applied LC and GSM to both randomly initialized and pretrained models, selecting the best results for each sparsity, noting the original works only applied these to pretrained models. Further, we experimented with STR (Kusupati et al., 2020), but omitted the results as we were unable to cover the exact sparsity range, being controllable only indirectly through regularization parameters. Detailed training and hyperparameters can be found in the corresponding subsection of Appendix A.2.

Incorporating SMS into BIMP, GMP, and DPF consistently improves performance, despite their deviation from IMP. BIMP benefits the most, likely due to the decaying learning rate facilitating convergence in each phase. In comparison, SMS stands out as a simple yet effective way to enhance the competitiveness of magnitude-based pruning methods, noting that SMS-enhanced methods increase computational costs by branching into $m$ models, a cost mitigable through parallelizing.

## 4 RELATED WORK

We review the related literature, focusing on sparsity-related studies. We refer to Hoefler et al. (2021) for a comprehensive review of sparsification approaches.

**Model Averaging.** *Stochastic Weight Averaging* (Izmailov et al., 2018) averages parameters across the SGD trajectory for improved generalization. Wortsman et al. (2022a) and Rame et al. (2022) demonstrate model soups' enhanced generalization and OOD-performance by averaging models finetuned with varying hyperparameters. The approach closest to ours in Section 3.3 is *Late-phase learning* (von Oswald et al., 2020), independently training and averaging specific parameters, albeit

without pruning. Gueta et al. (2023) explores fine-tuning in language models, revealing a clustering-like behavior with regions around close models containing potentially superior models. Croce et al. (2023) explore soups of adversarially-robust models, while Choshen et al. (2022) enhance base models by merging multiple finetuned ones. Wortsman et al. (2022b) introduce *robust fine-tuning* through averaging zero-shot and fine-tuned models. Similar to SMS, concurrent work by Jolicoeur-Martineau et al. (2023) regularly averages independently trained models, although without pruning.

We highlight key distinctions between our work and existing studies that combine sparsity with parameter averaging. Yin et al. (2022b) utilize dynamic sparse training, averaging models within a single run with fixed hyperparameters, in contrast to our prune-after-training method that averages models across multiple runs with diverse hyperparameter settings. Their prune-and-grow approach, exploring different sparsity patterns and requiring re-pruning to maintain sparsity, contrasts with our method which deliberately avoids re-pruning by keeping consistent sparsity patterns. We explicitly demonstrate that this approach significantly improves upon the re-pruning approach (IMP-RePrune), even when using strategies like CIA or CAA that are designed to mitigate the impact of re-pruning (Yin et al., 2022b). Similarly, Yin et al. (2022a) employ IMP with weight rewinding, averaging IMP subnetworks of different prune-retrain-cycles across a single training trajectory, which also requires re-pruning, unlike our approach of averaging parallely trained models. Furthermore, their objective is to generate lottery tickets, as opposed to creating sparse models for inference. In a similar vein, Stripelis et al. (2022) introduce FedSparsify, a Federated Learning algorithm that centrally updates a global mask with local client masks, resolving disparities through majority voting. Furthermore, our work is distinct from that of Jaiswal et al. (2023), who focus on averaging early pruning masks for mask generation, whereas we concentrate on averaging parameters of sparse models.

**Mode Connectivity.** Neyshabur et al. (2020) demonstrate that models trained from scratch are not linearly connected, while models finetuned from a pretrained model tend to be similar and reside within the same loss basin. Entezari et al. (2022) conjecture different-seed trained models are linear mode connected up to neuron permutations. Partially demonstrating this, Ainsworth et al. (2023) propose permutation algorithms for transforming models into a shared loss basin and Singh & Jaggi (2020) employ *model fusion* for neuron soft-alignment, further also enhancing filter pruning by fusing dense into sparse models. Similarly, Benzing et al. (2022) introduce a permutation algorithm, demonstrating that models share a loss valley (up to permutation) even at initialization. Jordan et al. (2022) explore 'variance collapse' in interpolations of deep networks, proposing mitigation strategies. Several works (Frankle et al., 2020; Evci et al., 2022; Paul et al., 2023) study IMP's stability to randomness, specifically with weight rewinding (IMP-WR). Evci et al. (2022) demonstrate that trained lottery tickets and IMP-WR solutions converge to identical basins, while Paul et al. (2023) find successive IMP-WR solutions at varied sparsity are linearly mode connected, maintaining loss stability along the linear interpolation between adjacent solutions.

**Prediction Ensembling.** A range of studies focus on prediction ensembling, where outputs of multiple models are averaged (Lakshminarayanan et al., 2017; Huang et al., 2017; Garipov et al., 2018; Mehrtash et al., 2020; Chandak et al., 2023). In the sparsity context, Liu et al. (2021) leverage DST for efficient generation of diverse ensemble candidates. Whitaker & Whitley (2022) form ensembles by randomly pruning and retraining model copies, while Kobayashi et al. (2022) finetune subnetworks of a pretrained model. We refer to Ganaie et al. (2021) for a survey of ensembling.

## 5 DISCUSSION

Efficient, high-performing sparse networks are crucial in resource-constrained environments. However, sparse models cannot easily leverage the benefits of parameter averaging. We addressed this issue proposing SMS, a technique that merges models while preserving sparsity, substantially enhancing IMP and outperforming multiple baselines. By integrating SMS into magnitude-pruning methods during training, we elevated their performance and competitiveness. Despite the focus on pruning, a single type of network compression, we think that our work serves as an important step towards understanding and improving sparsification algorithms.

ACKNOWLEDGEMENTS

This research was partially supported by the DFG Cluster of Excellence MATH+ (EXC-2046/1, project id 390685689) funded by the Deutsche Forschungsgemeinschaft (DFG). We would like to thank Berkant Turan and Christophe Roux for providing useful comments.

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

## APPENDICES

## A  TECHNICAL DETAILS AND TRAINING SETTINGS

### A.1  PRETRAINING

**Training settings and metrics.** Table 3 shows the exact pretraining settings for each dataset-architecture pair, reporting the number of epochs used for pretraining, the batch size, weight decay as well as the learning rate used. We stick to SGD as the optimizer, noting that a variety of other optimization methods for training deep neural networks exist (see e.g. Kingma & Ba, 2014; Pokutta et al., 2020). We keep momentum at the default value of 0.9. The last column reports the performance we achieve when performing regular dense training. For image classification tasks, we report the top-1 test accuracy being the fraction of correctly classified test samples. For semantic segmentation, we used pretrained backbones and evaluated the mean Intersection-over-Union (IoU) on the validation dataset, which we use as the test set. For NMT, we report the BLEU score on the test set with sequence length limited to 128. We utilized label smoothing and gradient clipping for MaxViT. If needed, we report the *theoretical speedup*, a metric indicating the FLOPs ratio for inference between dense and sparse models (Blalock et al., 2020). The speedup, defined as $F_d/F_s$ where $F_d$ and $F_s$ are the FLOPs required for dense and pruned models respectively, depends solely on the distribution of pruned weights, not on the values attained by non-zero parameters. FLOPs are computed using a single test batch, with code adapted from the *ShrinkBench* framework (Blalock et al., 2020).

Table 3: Exact pretraining configurations in our experiments.

| Dataset | Network (number of weights) | Epochs | Batch size | Weight decay | Learning rate ($t$ = training epoch) | Unpruned test accuracy/IoU/BLEU |
|---|---|---|---|---|---|---|
| CIFAR-10 | ResNet-18 (11 Mio) | 200 | 128 | 5e-4 | $\eta_t = \begin{cases} 0.1 & t \in [1, 90], \\ 0.01 & t \in [91, 180], \\ 0.001 & t \in [181, 200] \end{cases}$ | 95.0% ±0.04% |
| Celeb-A | ResNet-18 (11 Mio) | 100 | 256 | 1e-5 | linear from 0.1 to 0 | 98.9% ±0.01% |
| CIFAR-100 | WRN-20 (26 Mio) | 200 | 128 | 2e-4 | linear from 0.1 to 0 | 76.5% ±0.1% |
| ImageNet | ResNet-50 (26 Mio) | 90 | 256 | 1e-4 | linear from 0.1 to 0 | 76.12% ±0.01% |
| ImageNet | MaxViT (31 Mio) | 200 | 256 | 1e-5 | $\eta_t = \begin{cases} 0.2\frac{t}{20} & t \in [1, 20], \\ 0.2 & t \in [20, 60], \\ 0.02 & t \in [61, 120], \\ 0.002 & t \in [121, 160], \\ 0.0002 & t \in [161, 200] \end{cases}$ | 78.0% ±0.02% |
| CityScapes | PSPNet (68 Mio) | 300 | 12 | 1e-5 | $\eta_t = \begin{cases} 0.1\frac{t}{20} & t \in [1, 20], \\ 0.1 & t \in [20, 100], \\ 0.01 & t \in [101, 200], \\ 0.001 & t \in [201, 270], \\ 0.0001 & t \in [271, 290] \\ 0.00001 & t \in [291, 300] \end{cases}$ | 58.3 IoU ±0.5 |
| WMT16 (EN-DE) | T5-small (77 Mio) | 5 | 16 | 1e-5 | $\eta_t = \begin{cases} 0.1t & t \in [0, 1.0], \\ 0.1 & t \in [1, 2[, \\ 0.01 & t \in [2, 3[, \\ 0.001 & t \in [3, 4[, \\ 0.0001 & t \in [4, 5] \end{cases}$ | 24.56 BLEU ±0.007 |

### A.2  PRUNING AND RETRAINING

**Pruning settings.** Identifying which weights to remove is essential for successful magnitude pruning, with multiple methods developed to address this. Zhu & Gupta (2017) presented the UNIFORM allocation that prunes each layer to the same relative sparsity level. Gale et al. (2019) refined this approach to UNIFORM+, leaving the first convolutional layer dense and capping pruning in the final fully-connected layer at 80%. Evci et al. (2020) reformulate the Erdős-Rényi kernel (ERK) (Mocanu et al., 2018) to consider layer and kernel dimensions for layerwise sparsity distribution. Lee et al. (2020) suggested *Layer-Adaptive Magnitude-based Pruning* (LAMP), which minimizes output distortion at pruning time measured by the $L_2$-distortion on the worst-case input. Throughout this work, we stick to the GLOBAL allocation, in which all trainable parameters are treated as a single vector and a global threshold is computed to remove parameters independent of the layer they belong to. In experiments where we prune convolutional filters instead of weights, we adopt the $L_2$-norm criterion from Li et al. (2016), ensuring a uniform sparsity distribution across layers.

We follow the recommendations of Evci et al. (2020) and Dettmers & Zettlemoyer (2019) and refrain from pruning biases and batch-normalization parameters, as their negligible weight contribution is offset by their significant performance impact. Moreover, for GMP experiments, we opt for the global selection criterion due to its superior performance compared to UNIFORM+ (Zimmer et al., 2023).

**Retraining schedules.**     The choice of learning rate schedule during the retraining phase has recently attracted interest due to its significant influence on the performance of pruned networks. To avoid the undesired necessity of individually tuning the learning rate schedule in each phase, various retraining schedules have been devised to transpose the original learning rate schedule to the retraining phase. We briefly outline these schedules. Let $T$ represent the total epochs the original network is trained for with the learning rate schedule $(\eta_t)_{t \leq T}$, and let $T_{rt}$ denote the epochs allocated for retraining per prune-retrain cycle. The following retraining schedules have been proposed.

- FINE TUNING (FT, Han et al., 2015): Retrains the pruned network using the constant learning rate, $\eta_T$, from the last epoch of the original training.
- LEARNING RATE REWINDING (LRW, Renda et al., 2020): Utilizes the last $T - T_{rt}$ learning rates from the original training.
- SCALED LEARNING RATE RESTARTING (SLR, Le & Hua, 2021): Compresses the original learning rate schedule into the retraining timeframe with a short warm-up.
- CYCLIC LEARNING RATE RESTARTING (CLR, Le & Hua, 2021): Employs a cosine based schedule with a short warm-up to $\eta_1$.
- LINEAR LEARNING RATE RESTARTING (LLR, Zimmer et al., 2023): A linear decay from $\eta_1$ to zero during each retrain cycle.
- ADAPTIVE LINEAR LEARNING RATE RESTARTING (ALLR, Zimmer et al., 2023): LLR but dynamically adapts the initial learning rate based on the impact of the previous pruning step and the retraining time available, addressing both the length of a prune-retrain cycle and the performance drop induced by pruning.

**Hyperparameters for Retraining.**     The choice of hyperparameters during retraining significantly impacts the tradeoff between model performance and achieved sparsity. We briefly discuss the hyperparameters applied during the retraining phase, differentiating them from those we leave unchanged compared to the pretraining phase. The exact retraining hyperparameters are specified explicitly in the descriptions of each experiment or in the corresponding subsection in Appendix B.

Specifically, we retain the same batch size and weight decay parameters as used in pretraining. For each IMP-based experiment, we treat the retraining learning rate schedule, the number of retraining epochs, and the number of phases as experiment-specific hyperparameters.

- **Learning Rate Schedule:** The retraining schedule has a dramatic impact on the final performance, as outlined in the previous paragraph. Zimmer et al. (2023) demonstrate that LLR and ALLR surpass previously proposed methods across a broad spectrum of architectures, sparsity levels, and retraining durations. Thus, we adhere to these schedules in our experiments, specifying our choice explicitly when important.
- **Number of Retraining Epochs:** The number of epochs in retraining influences the extent to which the model can recover the pre-pruned accuracy. Further, in the high sparsity regime, sufficient retraining is required to ensure that models are averageable, as highlighted in Figure 4b.
- **Number of Phases:** The number of prune-retrain phases similarly impacts the performance vs. sparsity tradeoff. In general, high goal sparsity levels require multiple phases.

**Model Soups and Batch-Normalization statistics.**     Throughout our experiments, we also explored variants of *LearnedSoup* (Wortsman et al., 2022a), which learns the coefficients $\lambda_i$ to maximize the validation accuracy. Specifically, we observed improvements when utilizing knowledge distillation techniques (Hinton et al., 2015) with the original pretrained model as the teacher, instead of minimizing the validation loss as suggested by Wortsman et al. (2022b). Nevertheless, these improvements were marginal, so we opted for UniformSoup and GreedySoup for simplicity and to avoid introducing new hyperparameters.

Moreover, we noticed that in later IMP phases, assuming standard IMP rather than starting from an averaged model as in SMS, sparse connectivities tend to diverge, leading to diminished sparsity upon averaging. Specifically, while in the first phase two model splits converge to the same loss basin, subsequent pruning may project them into different subspaces, motivating the combination of

individual models into a single one at the end of each phase to ensure that effectively averaging them remains possible. Therefore, we also explored weight alignment, as notably proposed by Ainsworth et al. (2023) and Singh & Jaggi (2020) (see Section 4 for a detailed discussion), hoping to permute models in later phases to a common linear subspace. Although we partially mitigated the sparsity reduction, we were unable to fully recover the original sparsity. This approach may only work if different IMP runs share the same distribution of sparsity among layers (i.e., a specific layer in model one must have the same sparsity as the same layer in model two), which is generally not the case.

When constructing an averaged model or changing the parameters in any other way, the Batch-Normalization statistics have to be updated, which can be done by performing a forward pass on (part of) the training data without backpropagation. Since doing so only for model soups could potentially skew the results, we decided to recompute the statistics for single models as well to have a comparable setting independent of whether we changed the parameters or not. In particular, we enforce using the entire train data loader and we fixed its random batch ordering to ensure reproducibility and to avoid the batch ordering having any influence.

### A.3 PRUNING DURING TRAINING

**Hyperparameters for Pruning during Training algorithms.** Unless stated otherwise, we set weight decay to 1e-4 and momentum to 0.9, with all methods following a linear learning rate schedule starting from 1e-1. For GSM and LC, we select the best result either from scratch or using a pretrained model, applying these methods for 10, 20, or 40 epochs. When extending BIMP, GMP and DPF with SMS, we choose the number of copies to create within a phase, $m$, between 2 and 3. Further, we tuned the epoch at which we begin to train multiple copies between 50 and 75. Otherwise, we applied the following hyperparameter grids.

- BIMP
    - Initial training budget epochs: $60, 75$.
    - Number of pruning phases of equal length: $1, 2, 3$.
- GMP
    - Equally distributed pruning steps: $5, 9, 18, 45$.
- DPF
    - Equally distributed pruning steps: $9, 18$.
- GSM
    - Momentum: $0.9, 0.95$.
    - Weight decay: 1e-4, 1e-5.
- LC
    - Weight decay: 1e-4, 1e-5.
- DNW
    - Weight decay: 1e-4, 1e-5.
- DST
    - Weight decay: 1e-4, 1e-5.
    - $\alpha$: 1e-7, 5e-7, 1e-6, 2e-6, 5e-6, 8e-6, 1e-5, 1e-4.

# B  EXTENDED RESULTS

This section contains additional tables and plots. The subsections follow the same structure as the main experimental section. To maintain transparency, we explicitly mention the retraining schedule and duration in the captions of tables and figures, or in the beginning of the subsection if suitable.

## B.1  EVALUATING SPARSE MODEL SOUPS

Table 4: WideResNet-20 on CIFAR-100 (unstructured pruning): Test accuracy comparison of SMS to several baselines for target sparsities 90% (top) and 98% (bottom) given three prune-retrain cycles. We report results using UniformSoup as well as GreedySoup for merging, employing ALLR as the retraining schedule for 10 epochs of retraining per phase. Results are averaged over multiple seeds with standard deviation included. The best value is highlighted in bold.

**CIFAR-100** (90%)

| Accuracy of | Sparsity 53.6% (Phase 1) | | | Sparsity 78.5% (Phase 2) | | | Sparsity 90.0% (Phase 3) | | |
|---|---|---|---|---|---|---|---|---|---|
| | $m = 3$ | $m = 5$ | $m = 10$ | $m = 3$ | $m = 5$ | $m = 10$ | $m = 3$ | $m = 5$ | $m = 10$ |
| **SMS** (uniform) | **76.30 ±0.07** | **76.38 ±0.29** | **76.45 ±0.34** | **76.55 ±0.17** | **76.89 ±0.42** | **76.94 ±0.25** | **75.98 ±0.43** | 76.26 ±0.76 | **76.67 ±0.02** |
| best candidate | 76.03 ±0.06 | 76.18 ±0.03 | 76.13 ±0.35 | 75.88 ±0.49 | 75.78 ±0.35 | 75.82 ±0.21 | 75.15 ±0.05 | 75.23 ±0.38 | 75.52 ±0.30 |
| mean candidate | 75.88 ±0.00 | 75.86 ±0.15 | 75.93 ±0.27 | 75.50 ±0.21 | 75.48 ±0.12 | 75.45 ±0.13 | 74.99 ±0.22 | 75.06 ±0.42 | 75.15 ±0.16 |
| **SMS** (greedy) | 76.28 ±0.10 | 76.04 ±0.16 | 76.12 ±0.32 | 76.16 ±0.21 | 76.45 ±0.61 | 76.45 ±0.28 | 75.48 ±0.16 | 75.91 ±0.08 | 75.81 ±0.13 |
| best candidate | 76.05 ±0.03 | 76.18 ±0.03 | 76.11 ±0.33 | 75.44 ±0.11 | 75.66 ±0.19 | 75.59 ±0.35 | 75.14 ±0.06 | 75.08 ±0.35 | 74.97 ±0.11 |
| mean candidate | 75.93 ±0.02 | 75.86 ±0.15 | 75.92 ±0.26 | 75.26 ±0.11 | 75.34 ±0.15 | 75.24 ±0.19 | 74.87 ±0.01 | 74.81 ±0.23 | 74.72 ±0.21 |
| IMP$_{m\times}$ | 76.30 ±0.42 | 75.97 ±0.30 | 76.27 ±0.02 | 75.69 ±0.43 | 75.86 ±0.48 | 75.91 ±0.25 | 74.59 ±0.61 | 74.73 ±0.42 | 75.00 ±0.57 |
| IMP-RePrune | — N/A — | | | — N/A — | | | 75.70 ±0.59 | 75.68 ±0.25 | 75.60 ±0.23 |
| IMP | — 75.64 ±0.21 — | | | — 75.51 ±0.52 — | | | — 74.91 ±0.71 — | | |

**CIFAR-100** (98%)

| Accuracy of | Sparsity 72.8% (Phase 1) | | | Sparsity 92.6% (Phase 2) | | | Sparsity 98.0% (Phase 3) | | |
|---|---|---|---|---|---|---|---|---|---|
| | $m = 3$ | $m = 5$ | $m = 10$ | $m = 3$ | $m = 5$ | $m = 10$ | $m = 3$ | $m = 5$ | $m = 10$ |
| **SMS** (uniform) | **76.50 ±0.16** | **76.59 ±0.13** | **76.75 ±0.28** | **75.55 ±0.60** | **76.19 ±0.37** | **76.21 ±0.43** | **72.67 ±0.29** | **72.90 ±0.64** | **73.05 ±0.45** |
| best candidate | 75.58 ±0.19 | 75.71 ±0.08 | 75.96 ±0.13 | 74.51 ±0.47 | 75.01 ±0.74 | 75.00 ±0.34 | 71.77 ±0.04 | 71.77 ±0.37 | 72.21 ±0.02 |
| mean candidate | 75.37 ±0.12 | 75.58 ±0.03 | 75.55 ±0.26 | 74.32 ±0.40 | 74.71 ±0.48 | 74.70 ±0.42 | 71.41 ±0.09 | 71.61 ±0.40 | 71.66 ±0.19 |
| **SMS** (greedy) | 76.06 ±0.69 | 76.43 ±0.24 | 76.60 ±0.47 | 75.34 ±0.15 | 75.39 ±0.44 | 75.51 ±0.66 | 72.08 ±0.23 | 71.86 ±0.64 | 72.44 ±0.20 |
| best candidate | 75.58 ±0.19 | 75.65 ±0.00 | 75.94 ±0.15 | 74.85 ±0.04 | 74.53 ±0.42 | 74.57 ±0.21 | 71.05 ±0.43 | 71.01 ±0.49 | 71.47 ±0.23 |
| mean candidate | 75.37 ±0.12 | 75.54 ±0.03 | 75.54 ±0.27 | 74.52 ±0.25 | 74.27 ±0.52 | 74.20 ±0.31 | 70.84 ±0.41 | 70.69 ±0.75 | 70.87 ±0.01 |
| IMP$_{m\times}$ | 75.85 ±0.26 | 76.05 ±0.00 | 75.76 ±0.24 | 74.09 ±0.24 | 74.19 ±0.44 | 74.74 ±0.06 | 70.92 ±0.07 | 70.31 ±0.52 | 71.85 ±0.15 |
| IMP-RePrune | — N/A — | | | — N/A — | | | 68.19 ±0.44 | 65.53 ±0.06 | 63.62 ±0.90 |
| IMP | — 75.54 ±0.41 — | | | — 74.09 ±0.13 — | | | — 70.74 ±0.08 — | | |

Table 5: ResNet-18 on CIFAR-10 (unstructured pruning): Test accuracy comparison of SMS to several baselines for target sparsity 98% given three prune-retrain cycles. We report results using UniformSoup as well as GreedySoup for merging, employing ALLR as the retraining schedule for 20 epochs of retraining per phase. Results are averaged over multiple seeds with standard deviation included. The best value is highlighted in bold.

**CIFAR-10** (98%)

| Accuracy of | Sparsity 72.8% (Phase 1) | | | Sparsity 92.6% (Phase 2) | | | Sparsity 98.0% (Phase 3) | | |
|---|---|---|---|---|---|---|---|---|---|
| | $m = 3$ | $m = 5$ | $m = 10$ | $m = 3$ | $m = 5$ | $m = 10$ | $m = 3$ | $m = 5$ | $m = 10$ |
| **SMS** (uniform) | 95.41 ±0.00 | **95.46 ±0.13** | **95.64 ±0.01** | 95.45 ±0.01 | **95.59 ±0.09** | **95.61 ±0.03** | 94.91 ±0.16 | **95.24 ±0.11** | **95.40 ±0.10** |
| best candidate | 94.84 ±0.10 | 94.96 ±0.13 | 94.94 ±0.07 | 94.92 ±0.07 | 95.11 ±0.06 | 95.12 ±0.02 | 94.40 ±0.02 | 94.69 ±0.16 | 94.81 ±0.09 |
| mean candidate | 94.70 ±0.13 | 94.82 ±0.00 | 94.75 ±0.08 | 94.85 ±0.04 | 94.93 ±0.03 | 94.94 ±0.01 | 94.32 ±0.12 | 94.57 ±0.16 | 94.66 ±0.13 |
| **SMS** (greedy) | **95.42 ±0.04** | 95.45 ±0.18 | 95.50 ±0.05 | **95.46 ±0.08** | 95.27 ±0.24 | 95.32 ±0.20 | **95.01 ±0.08** | 94.92 ±0.04 | 94.94 ±0.25 |
| best candidate | 94.84 ±0.10 | 94.96 ±0.13 | 94.94 ±0.07 | 94.96 ±0.26 | 94.86 ±0.09 | 94.92 ±0.08 | 94.55 ±0.11 | 94.48 ±0.01 | 94.54 ±0.06 |
| mean candidate | 94.70 ±0.13 | 94.82 ±0.00 | 94.75 ±0.08 | 94.80 ±0.15 | 94.76 ±0.11 | 94.79 ±0.07 | 94.44 ±0.06 | 94.34 ±0.01 | 94.29 ±0.06 |
| IMP$_{m\times}$ | 95.18 ±0.08 | 95.16 ±0.16 | 95.19 ±0.18 | 95.02 ±0.11 | 95.11 ±0.18 | 95.20 ±0.02 | 94.62 ±0.28 | 94.61 ±0.02 | 94.59 ±0.23 |
| IMP-RePrune | — N/A — | | | — N/A — | | | 94.44 ±0.28 | 94.24 ±0.13 | 93.62 ±0.16 |
| IMP | — 94.71 ±0.08 — | | | — 94.92 ±0.01 — | | | — 94.17 ±0.04 — | | |

Table 6: ResNet-18 on CIFAR-10 (unstructured pruning): Test accuracy comparison of SMS to several baselines for target sparsities 80%, 90%, 95% in the One Shot setting, using ALLR for a retrain length of 20 epochs per phase. We report results using UniformSoup as well as GreedySoup for merging. Results are averaged over multiple seeds with standard deviation included. The best value is highlighted in bold.

**CIFAR-10**

| Accuracy of | Sparsity 80.0% (One Shot) $m = 3$ | Sparsity 90.0% (One Shot) $m = 3$ | Sparsity 95.0% (One Shot) $m = 3$ |
|---|---|---|---|
| SMS (uniform) | **95.49 ±0.05** | **95.42 ±0.06** | 95.03 ±0.15 |
| best candidate | 95.04 ±0.06 | 94.84 ±0.03 | 94.63 ±0.06 |
| mean candidate | 94.92 ±0.08 | 94.76 ±0.03 | 94.48 ±0.05 |
| SMS (greedy) | 95.36 ±0.16 | 95.41 ±0.07 | **95.04 ±0.19** |
| best candidate | 95.04 ±0.06 | 94.84 ±0.03 | 94.63 ±0.06 |
| mean candidate | 94.92 ±0.08 | 94.76 ±0.03 | 94.48 ±0.05 |
| IMP$_{m\times}$ | 95.17 ±0.17 | 95.02 ±0.01 | 94.72 ±0.24 |
| IMP | 95.02 ±0.05 | 94.71 ±0.28 | 94.38 ±0.02 |

Table 7: MaxViT on ImageNet (unstructured pruning): Test accuracy comparison of SMS to several baselines for target sparsities 75%, 80%, 85% in the One Shot setting, using ALLR for a retrain length of 10 epochs per phase. We report results using UniformSoup as well as GreedySoup for merging. Results are averaged over multiple seeds with standard deviation included. The best value is highlighted in bold.

**ImageNet**

| Accuracy of | Sparsity 75.0% (One Shot) $m = 3$ | Sparsity 80.0% (One Shot) $m = 3$ | Sparsity 85.0% (One Shot) $m = 3$ |
|---|---|---|---|
| **SMS** (uniform) | **78.31 ±0.21** | **78.12 ±0.17** | 77.59 ±0.24 |
| best candidate | 78.11 ±0.07 | 77.85 ±0.10 | 77.37 ±0.02 |
| mean candidate | 77.83 ±0.01 | 77.67 ±0.07 | 77.17 ±0.04 |
| **SMS** (greedy) | 78.21 ±0.14 | 78.06 ±0.01 | 77.45 ±0.13 |
| best candidate | 78.11 ±0.07 | 77.85 ±0.10 | 77.37 ±0.02 |
| mean candidate | 77.83 ±0.01 | 77.66 ±0.06 | 77.17 ±0.04 |
| IMP$_{m\times}$ | 78.17 ±0.15 | 77.99 ±0.12 | **77.68 ±0.23** |
| IMP | 78.07 ±0.09 | 77.88 ±0.06 | 77.34 ±0.12 |

Table 8: PSPNet on Cityscapes (unstructured pruning): Test accuracy comparison of SMS to several baselines for target sparsity 90% given two prune-retrain cycles of 50 retraining epochs each. We report results using UniformSoup as well as GreedySoup merging. Results are averaged over multiple seeds with standard deviation included. The best value is highlighted in bold.

**CityScapes** (90%)

| Accuracy of | Sparsity 68.3% (Phase 1) $m = 3$ | Sparsity 68.3% (Phase 1) $m = 5$ | Sparsity 90.0% (Phase 2) $m = 3$ | Sparsity 90.0% (Phase 2) $m = 5$ |
|---|---|---|---|---|
| **SMS** (uniform) | **58.52 ±0.15** | 58.47 ±0.10 | 58.73 ±0.20 | 58.40 ±0.30 |
| best candidate | 58.20 ±0.37 | 58.25 ±0.48 | 58.62 ±0.60 | 57.92 ±0.36 |
| mean candidate | 57.96 ±0.29 | 57.80 ±0.33 | 58.38 ±0.41 | 57.62 ±0.48 |
| **SMS** (greedy) | 58.14 ±0.14 | **58.63 ±0.36** | 58.46 ±0.24 | 58.79 ±0.09 |
| best candidate | 58.26 ±0.06 | 58.59 ±0.14 | 58.13 ±0.27 | 58.73 ±0.06 |
| mean candidate | 57.82 ±0.13 | 58.17 ±0.14 | 57.90 ±0.09 | 58.15 ±0.04 |
| IMP$_{m\times}$ | 57.39 ±0.06 | 58.35 ±0.08 | 58.27 ±0.42 | 58.39 ±0.17 |
| IMP-RePrune | — N/A — | | 58.10 ±0.24 | 58.64 ±0.34 |
| IMP | — 57.92 ±0.04 — | | — **58.89 ±0.23** — | |

Table 9: PSPNet on Cityscapes (unstructured pruning): Test accuracy comparison of SMS to several baselines for target sparsities 60%, 70%, 80% and 90% in the One Shot setting, using LLR as the retraining schedule for 50 epochs of retraining. We report results using UniformSoup as well as GreedySoup merging. Results are averaged over multiple seeds with standard deviation included. The best value is highlighted in bold.

**CityScapes**

| Accuracy of | Sparsity 60.0% (One Shot) $m = 3$ | Sparsity 70.0% (One Shot) $m = 3$ | Sparsity 80.0% (One Shot) $m = 3$ | Sparsity 90.0% (One Shot) $m = 3$ |
|---|---|---|---|---|
| **SMS** (uniform) | 58.11 ±0.22 | 57.59 ±0.38 | **58.40 ±0.03** | 58.19 ±0.28 |
| best candidate | 57.74 ±0.11 | 57.36 ±0.43 | 57.97 ±0.06 | 57.87 ±0.49 |
| mean candidate | 57.37 ±0.47 | 57.04 ±0.48 | 57.88 ±0.03 | 57.70 ±0.44 |
| **SMS** (greedy) | **58.41 ±0.13** | **58.13 ±0.31** | 57.95 ±0.40 | 57.30 ±0.21 |
| best candidate | 58.16 ±0.49 | 57.78 ±0.13 | 57.78 ±0.26 | 57.32 ±0.28 |
| mean candidate | 58.05 ±0.50 | 57.55 ±0.07 | 57.49 ±0.38 | 57.18 ±0.25 |
| $\text{IMP}_{m\times}$ | 58.02 ±0.09 | 58.09 ±1.04 | 58.26 ±0.13 | **58.47 ±0.22** |
| IMP | 57.44 ±0.71 | 58.05 ±0.28 | 57.43 ±0.24 | 56.99 ±0.69 |

Table 10: T5 on WMT16 (unstructured pruning): BLEU score comparison of SMS to several baselines for target sparsities 50%, 60%, 70% in the One Shot setting. We report results using UniformSoup as well as GreedySoup merging, employing ALLR as the retraining schedule for 2 epochs of retraining per phase. Results are averaged over multiple seeds with standard deviation included. The best value is highlighted in bold.

**WMT-16**

| Accuracy of | Sparsity 50.0% (One Shot) $m = 3$ | Sparsity 60.0% (One Shot) $m = 3$ | Sparsity 70.0% (One Shot) $m = 3$ |
|---|---|---|---|
| SMS (uniform) | 25.47 ±0.52 | **25.09 ±0.00** | **24.51 ±0.43** |
| best candidate | 25.39 ±0.03 | 24.96 ±0.26 | 24.12 ±0.01 |
| mean candidate | 25.16 ±0.08 | 24.79 ±0.19 | 24.03 ±0.01 |
| SMS (greedy) | **25.51 ±0.28** | 24.92 ±0.47 | 24.14 ±0.02 |
| best candidate | 25.39 ±0.03 | 24.96 ±0.26 | 24.12 ±0.01 |
| mean candidate | 25.16 ±0.08 | 24.79 ±0.19 | 24.03 ±0.01 |
| $\text{IMP}_{m\times}$ | 25.36 ±0.12 | 25.09 ±0.05 | 24.00 ±0.04 |
| IMP | 25.15 ±0.20 | 24.90 ±0.20 | 24.04 ±0.28 |

Table 11: WideResNet-20 on CIFAR-100 (structured pruning): Test accuracy comparison of SMS to several baselines for target sparsities 60% (top) and 80% (bottom) given three prune-retrain cycles. We report results using UniformSoup as well as GreedySoup for merging, employing ALLR as the retraining schedule for 10 epochs of retraining per phase. Results are averaged over multiple seeds with standard deviation included. The best value is highlighted in bold.

**CIFAR-100** (60%)

| Accuracy of | Sparsity 26.2% (Phase 1) | | | Sparsity 45.6% (Phase 2) | | | Sparsity 60.0% (Phase 3) | | |
| --- | --- | --- | --- | --- | --- | --- | --- | --- | --- |
| | $m=3$ | $m=5$ | $m=10$ | $m=3$ | $m=5$ | $m=10$ | $m=3$ | $m=5$ | $m=10$ |
| SMS (uniform) | 76.54 ±0.19 | **76.56 ±0.21** | **76.82 ±0.19** | 75.99 ±0.07 | **76.23 ±0.35** | **76.46 ±0.16** | **75.34 ±0.11** | 75.44 ±0.11 | **75.74 ±0.18** |
| best candidate | 75.12 ±0.04 | 75.19 ±0.06 | 75.23 ±0.07 | 74.73 ±0.15 | 74.90 ±0.12 | 74.85 ±0.16 | 74.31 ±0.42 | 74.45 ±0.11 | 74.53 ±0.16 |
| mean candidate | 74.98 ±0.12 | 74.87 ±0.06 | 74.86 ±0.05 | 74.49 ±0.14 | 74.53 ±0.14 | 74.53 ±0.06 | 73.84 ±0.36 | 73.95 ±0.19 | 73.95 ±0.05 |
| SMS (greedy) | **76.55 ±0.41** | 76.45 ±0.24 | 76.40 ±0.18 | 75.85 ±0.06 | 76.10 ±0.38 | 76.03 ±0.43 | 75.10 ±0.12 | 75.24 ±0.20 | 74.48 ±0.42 |
| best candidate | 75.09 ±0.01 | 75.19 ±0.06 | 75.21 ±0.10 | 74.88 ±0.16 | 74.92 ±0.11 | 74.80 ±0.23 | 74.04 ±0.15 | 74.28 ±0.37 | 74.23 ±0.22 |
| mean candidate | 74.96 ±0.09 | 74.87 ±0.06 | 74.88 ±0.02 | 74.62 ±0.12 | 74.56 ±0.28 | 74.43 ±0.20 | 73.85 ±0.05 | 73.78 ±0.12 | 73.87 ±0.14 |
| IMP$_{m\times}$ | 75.55 ±0.04 | | 75.92 ±0.16 | 74.84 ±0.02 | 75.12 ±0.09 | 75.68 ±0.47 | 74.28 ±0.05 | 74.63 ±0.32 | 74.73 ±0.94 |
| IMP-RePrune | | — N/A — | | | — N/A — | | 74.65 ±0.63 | **75.54 ±0.28** | 75.49 ±0.33 |
| IMP | | — 74.96 ±0.20 — | | | — 74.09 ±0.05 — | | | — 73.47 ±0.04 — | |

**CIFAR-100** (80%)

| Accuracy of | Sparsity 41.5% (Phase 1) | | | Sparsity 65.8% (Phase 2) | | | Sparsity 80.0% (Phase 3) | | |
| --- | --- | --- | --- | --- | --- | --- | --- | --- | --- |
| | $m=3$ | $m=5$ | $m=10$ | $m=3$ | $m=5$ | $m=10$ | $m=3$ | $m=5$ | $m=10$ |
| SMS (uniform) | **75.94 ±0.01** | **75.99 ±0.40** | **76.19 ±0.40** | **74.18 ±0.02** | **74.23 ±0.27** | **74.76 ±0.06** | 71.56 ±0.16 | 71.59 ±0.14 | 71.78 ±0.25 |
| best candidate | 74.65 ±0.29 | 74.65 ±0.11 | 74.78 ±0.17 | 73.27 ±0.26 | 73.22 ±0.27 | 73.71 ±0.18 | 70.61 ±0.11 | 70.58 ±0.50 | 70.96 ±0.33 |
| mean candidate | 74.44 ±0.17 | 74.37 ±0.16 | 74.39 ±0.13 | 72.90 ±0.08 | 72.94 ±0.23 | 73.19 ±0.09 | 70.50 ±0.08 | 70.31 ±0.52 | 70.40 ±0.23 |
| SMS (greedy) | 75.87 ±0.10 | 75.97 ±0.44 | 76.14 ±0.52 | 74.13 ±0.23 | 74.21 ±0.10 | 74.48 ±0.38 | 71.63 ±0.25 | 71.70 ±0.27 | 71.06 ±0.87 |
| best candidate | 74.70 ±0.23 | 74.65 ±0.11 | 74.80 ±0.15 | 73.18 ±0.22 | 73.40 ±0.32 | 73.59 ±0.25 | 70.86 ±0.30 | 70.93 ±0.04 | 70.37 ±0.79 |
| mean candidate | 74.47 ±0.13 | 74.37 ±0.16 | 74.38 ±0.14 | 72.95 ±0.33 | 73.03 ±0.07 | 73.13 ±0.10 | 70.56 ±0.45 | 70.37 ±0.04 | 69.83 ±0.88 |
| IMP$_{m\times}$ | 75.06 ±0.04 | | 75.39 ±0.09 | 73.44 ±0.22 | 73.70 ±0.06 | 73.96 ±0.54 | **72.06 ±0.10** | **72.15 ±0.58** | **72.32 ±0.58** |
| IMP-RePrune | | — N/A — | | | — N/A — | | 70.52 ±0.18 | 68.60 ±2.94 | 69.89 ±1.34 |
| IMP | | — 73.95 ±0.08 — | | | — 72.71 ±0.15 — | | | — 69.88 ±0.50 — | |

Table 12: ResNet-18 on CIFAR-10 (structured pruning): Test accuracy comparison of SMS to several baselines for target sparsities 40%, 50%, 60% in the One Shot setting, using ALLR as the retrain schedule for a retrain length of 20 epochs. We report results using UniformSoup as well as GreedySoup for merging. Results are averaged over multiple seeds with standard deviation included. The best value is highlighted in bold.

**CIFAR-10**

| Accuracy of | Sparsity 40.0% (One Shot) | Sparsity 50.0% (One Shot) | Sparsity 60.0% (One Shot) |
| --- | --- | --- | --- |
| | $m=3$ | $m=3$ | $m=3$ |
| SMS (uniform) | **94.91 ±0.00** | 94.68 ±0.02 | 94.32 ±0.10 |
| best candidate | 94.42 ±0.04 | 94.28 ±0.04 | 93.81 ±0.06 |
| mean candidate | 94.31 ±0.01 | 94.16 ±0.03 | 93.77 ±0.06 |
| SMS (greedy) | 94.87 ±0.02 | **94.70 ±0.08** | 94.31 ±0.11 |
| best candidate | 94.42 ±0.04 | 94.28 ±0.01 | 93.81 ±0.06 |
| mean candidate | 94.31 ±0.01 | 94.17 ±0.04 | 93.77 ±0.06 |
| IMP$_{m\times}$ | 94.77 ±0.01 | 94.38 ±0.23 | **94.34 ±0.16** |
| IMP | 94.45 ±0.18 | 94.06 ±0.17 | 93.96 ±0.34 |

## B.2   EXAMINING SPARSE MODEL MERGING

### B.2.1   EXPLORING PARAMETERS BEYOND RANDOM SEEDS.

Similar to Figure 3, Figure 5 visualizes the effects of different hyperparameters given One Shot IMP (90% sparsity) with WRN-20 trained on CIFAR-100. Again, we created eight variations for each parameter (random seed, weight decay strength, retraining duration, and initial learning rate in a linear decay schedule) based on equidistant values around the defaults. The scatter plot compares the test accuracy of models in the uniform soup setting (averaged from pairs of models) against the maximum test accuracy within each pair.

To generate the averaging candidates, we employed the following hyperparameter configurations. For ImageNet, as showcased in Figure 3, our base configuration utilized ALLR for 5 retraining epochs with weight decay as stated in Table 3. While varying the random seed, we maintained the base configuration and selected eight distinct random seeds. In adjusting the weight decay, we adhered to the base configuration and experimented with weight decay strengths of 4e-5, 6e-5, 8e-5, 1e-4, 1.2e-4, 1.4e-4, 1.6e-4, 1.8e-4. For retraining length variation, we examined all integral values between 2 and 9 epochs. In terms of retraining schedule modification, we adopted a linearly decaying learning rate schedule, tuning the initial value among 2e-2, 4e-2, 6e-2, 8e-2, 1e-1, 1.2e-1, 1.4e-1 and 1.6e-1. For CIFAR-100, as in Figure 5, we used ALLR for 10 epochs with weight decay as in Table 3. To adjust the weight decay, we experimented with weight decay strengths of 1e-4, 2e-4, 3e-4, 4e-4, 5e-4, 6e-4, 7e-4, 8e-4. For retraining length variation, we examined all integral values between 6 and 13 epochs. In terms of retraining schedule modification, we adopted a linearly decaying learning rate schedule, tuning the initial value among 6e-2, 7e-2, 8e-2, 9e-2, 1e-1, 1.1e-1, 1.2e-1 and 1.3e-1.

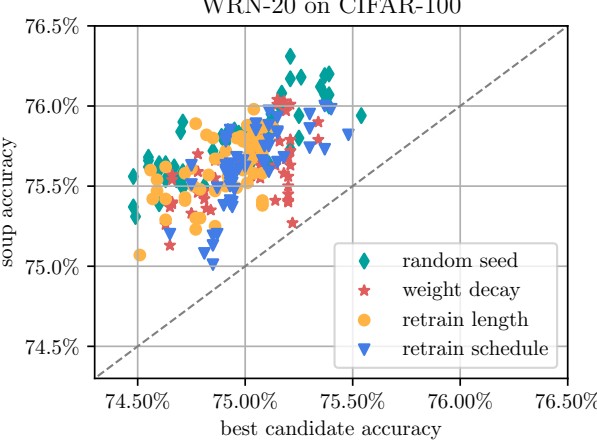

Figure 5: WideResNet-20 on CIFAR-100: Accuracy of average of two models vs. the maximal individual accuracy. All models are pruned to 90% sparsity (One Shot) and retrained, varying the indicated hyperparameters.

### B.2.2 INSTABILITY TO RANDOMNESS AND RECOVERING IT.

Figure 6 replicates the plots from Figure 4, albeit for CIFAR-10, adhering to the identical retraining hyperparameter configuration delineated in the main text.

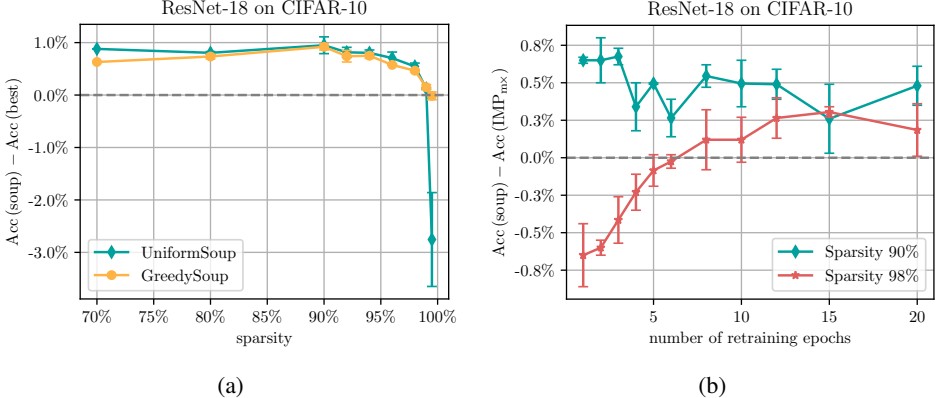

(a)                                                                   (b)

Figure 6: ResNet-18 on CIFAR-10: (a) Accuracy difference between the soup and best performing model after One Shot pruning and retraining. The lines for UniformSoup and GreedySoup show the envelope considering all retraining schedules and durations. (b) Accuracy difference between the soup ($m = 3$) and IMP$_{3\times}$ retrained three times as long as indicated on the x-axis, using One Shot pruning to 90% and 98% sparsity. Results are averaged over multiple random seeds with min-max bands indicated.

### B.2.3    OOD-ROBUSTNESS OF SPARSE MODEL SOUPS.

Figure 7 depicts OOD-robustness effects for One Shot IMP on WRN-20 trained on CIFAR-100 at 90% sparsity, while Figure 8 does the same for ResNet-50 on ImageNet at 70% sparsity. For each parameter (random seed, weight decay strength, retraining duration, and initial learning rate in a linear decay schedule), we formulated eight variations centered on default values (see subsubsection B.2.1 for exact hyperparameters). Contrary to previous scatter plots, these evaluate models on the robustness benchmarks CIFAR-100-C and Imagenet-C. The plots contrast the OOD accuracy in the uniform soup setting (averaged across model pairs) with the peak OOD accuracy of each pair. The OOD accuracy is computed on the entire corrupted dataset, i.e., among all corruption types and degrees of severity (ranging from 1 to 5).

Further, Table 13 and Table 14 display the OOD-robustness evaluated on CIFAR-100-C and ImageNet-C, respectively, when aiming for higher sparsities and using multiple cycles. SMS consistently improves over the baselines and improves the out-of-distribution accuracy significantly. For the pretrained models we obtain a base ood accuracy of $49.03\%(\pm0.33)$ for CIFAR-100-C and $40.35\%(\pm0.30)$ for ImageNet-C.

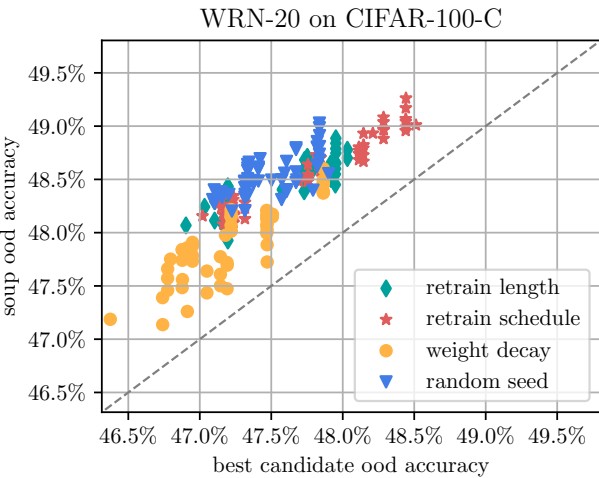

Figure 7: WideResNet-20 evaluated on CIFAR-100-C: OOD Accuracy of average of two models vs. the maximal individual OOD accuracy. All models are pruned to 90% sparsity (One Shot) and retrained, varying the indicated hyperparameters.

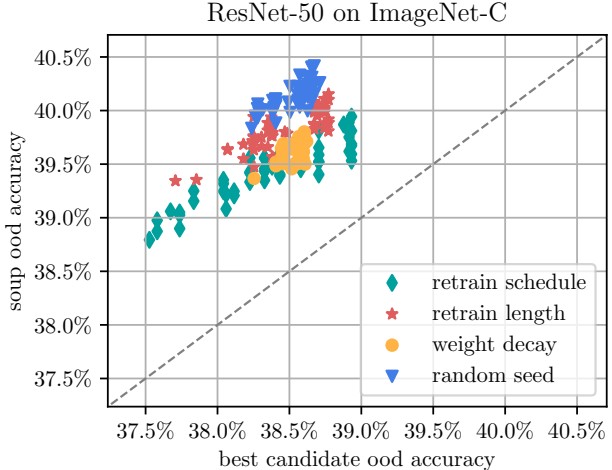

Figure 8: ResNet-50 evaluated on ImageNet-C: OOD Accuracy of average of two models vs. the maximal individual OOD accuracy. All models are pruned to 70% sparsity (One Shot) and retrained, varying the indicated hyperparameters.

Table 13: WideResNet-20 trained on CIFAR-100 and evaluated on CIFAR-100-C (unstructured pruning): OOD accuracy comparison of SMS to several baselines for target sparsities 90% (top) and 98% (bottom) given three prune-retrain cycles. We only report results using UniformSoup for merging, employing ALLR as the retraining schedule for 10 epochs of retraining per phase. Results are averaged over multiple seeds with standard deviation included. The best value is highlighted in bold.

**CIFAR-100** (90%)

| Accuracy of | Sparsity 53.6% (Phase 1) | | | Sparsity 78.5% (Phase 2) | | | Sparsity 90.0% (Phase 3) | | |
|---|---|---|---|---|---|---|---|---|---|
| | $m=3$ | $m=5$ | $m=10$ | $m=3$ | $m=5$ | $m=10$ | $m=3$ | $m=5$ | $m=10$ |
| **SMS** | **49.60 ±0.14** | **49.48 ±0.20** | **49.57 ±0.24** | **49.90 ±0.11** | **50.09 ±0.08** | **50.33 ±0.14** | 48.65 ±0.31 | 48.86 ±0.06 | **49.38 ±0.14** |
| best candidate | 49.31 ±0.14 | 49.18 ±0.22 | 49.17 ±0.23 | 48.51 ±0.09 | 48.64 ±0.08 | 48.82 ±0.17 | 47.42 ±0.29 | 47.87 ±0.15 | 47.87 ±0.09 |
| mean candidate | 49.10 ±0.18 | 48.97 ±0.15 | 48.97 ±0.18 | 48.40 ±0.12 | 48.33 ±0.11 | 48.37 ±0.03 | 47.29 ±0.28 | 47.51 ±0.20 | 47.61 ±0.07 |
| $IMP_{m\times}$ | 49.25 ±0.10 | 49.01 ±0.52 | 49.17 ±0.38 | 48.06 ±0.07 | 48.24 ±0.52 | 48.17 ±0.65 | 47.10 ±0.74 | 46.59 ±0.05 | 47.12 ±0.52 |
| IMP-RePrune | — N/A — | | | — N/A — | | | **48.72 ±0.25** | **49.03 ±0.15** | 49.35 ±0.11 |
| IMP | — 49.41 ±0.22 — | | | — 48.25 ±0.08 — | | | — 46.88 ±0.27 — | | |

**CIFAR-100** (98%)

| Accuracy of | Sparsity 72.8% (Phase 1) | | | Sparsity 92.6% (Phase 2) | | | Sparsity 98.0% (Phase 3) | | |
|---|---|---|---|---|---|---|---|---|---|
| | $m=3$ | $m=5$ | $m=10$ | $m=3$ | $m=5$ | $m=10$ | $m=3$ | $m=5$ | $m=10$ |
| **SMS** | **50.05 ±0.09** | **50.01 ±0.07** | **50.28 ±0.09** | **48.30 ±0.15** | **48.52 ±0.07** | **48.80 ±0.12** | **44.18 ±0.43** | **44.81 ±0.26** | **44.80 ±0.75** |
| best candidate | 49.36 ±0.17 | 49.01 ±0.16 | 49.18 ±0.02 | 47.11 ±0.12 | 47.07 ±0.20 | 47.27 ±0.25 | 43.37 ±0.41 | 43.49 ±0.23 | 43.72 ±0.64 |
| mean candidate | 48.95 ±0.07 | 48.68 ±0.09 | 48.67 ±0.09 | 46.87 ±0.14 | 46.80 ±0.08 | 46.87 ±0.23 | 42.94 ±0.52 | 43.31 ±0.34 | 43.17 ±0.54 |
| $IMP_{m\times}$ | 48.70 ±0.19 | 48.81 ±0.10 | 48.66 ±0.17 | 46.20 ±0.27 | 45.90 ±0.10 | 46.17 ±0.07 | 41.97 ±0.17 | 41.62 ±1.55 | 42.76 ±0.30 |
| IMP-RePrune | — N/A — | | | — N/A — | | | 39.57 ±0.82 | 37.17 ±1.04 | 35.28 ±1.26 |
| IMP | — 48.60 ±0.14 — | | | — 45.89 ±0.14 — | | | — 42.43 ±0.58 — | | |

Table 14: ResNet-50 trained on ImageNet and evaluated on ImageNet-C (unstructured pruning): OOD accuracy comparison of SMS to several baselines for target sparsity 90% given three prune-retrain cycles. We only report results using UniformSoup for merging, employing ALLR as the retraining schedule for 10 epochs of retraining per phase. Results are averaged over multiple seeds with standard deviation included. The best value is highlighted in bold.

**ImageNet** (90%)

| Accuracy of | Sparsity 53.6% (Phase 1) | | | Sparsity 78.5% (Phase 2) | | | Sparsity 90.0% (Phase 3) | | |
|---|---|---|---|---|---|---|---|---|---|
| | $m=3$ | $m=5$ | $m=10$ | $m=3$ | $m=5$ | $m=10$ | $m=3$ | $m=5$ | $m=10$ |
| **SMS** | **42.19 ±0.17** | **42.65 ±0.17** | **42.94 ±0.02** | **41.17 ±0.15** | **41.62 ±0.10** | **42.11 ±0.06** | **38.70 ±0.02** | **39.23 ±0.15** | **39.70 ±0.02** |
| best candidate | 39.98 ±0.12 | 40.00 ±0.25 | 40.11 ±0.05 | 39.31 ±0.11 | 39.43 ±0.03 | 39.66 ±0.05 | 37.30 ±0.01 | 37.57 ±0.13 | 37.84 ±0.11 |
| mean candidate | 39.87 ±0.10 | 39.90 ±0.22 | 39.91 ±0.09 | 39.19 ±0.10 | 39.28 ±0.05 | 39.47 ±0.01 | 37.21 ±0.01 | 37.39 ±0.14 | 37.62 ±0.03 |
| $IMP_{m\times}$ | 40.05 ±0.17 | 40.36 ±0.29 | 40.44 ±0.11 | 39.31 ±0.02 | 39.46 ±0.05 | 39.45 ±0.28 | 37.13 ±0.22 | 37.47 ±0.26 | 37.36 ±0.34 |
| IMP-RePrune | — N/A — | | | — N/A — | | | 37.10 ±0.15 | 36.81 ±0.18 | 36.48 ±0.66 |
| IMP | — 39.84 ±0.05 — | | | — 38.74 ±0.02 — | | | — 36.64 ±0.00 — | | |

### B.2.4 REDUCING COMPRESSION-INDUCED UNFAIRNESS.

Classification model performance, usually measured by the top-1 accuracy, can mask the disproportionate effect of compression on individual class performance (Hooker et al., 2019; 2020; Paganini, 2020). Pruning often sacrifices difficult samples, benefiting well-performing classes and worsening the performance of others (Tran et al., 2022). Recent research by Ko et al. (2023) highlights the benefits of prediction ensembling in enhancing fairness metrics, including minority group performance.

We investigate whether model averaging mitigates pruning's adverse effects on fairness using ResNet-18 trained on the Celeb-A facial attribute recognition dataset, a fairness benchmark due to its strong sub-group and target label correlation. Table 15 compares the dense model, One Shot SMS, and IMP at sparsities of 90%, 95%, and 97%, where 20 epochs of retraining yields similar top-1 test accuracies for SMS and IMP. We report the recall for the disjoint sub-groups: Male-Blonde (MB), Male-Non-Blonde (MN), Female-Blonde (FB), and Female-Non-Blonde (FN), in addition to top-1 test accuracy. Although SMS and IMP reach similar test accuracy, differences in subgroup performance arise for sparsities above 90%. SMS notably increases recall for MB and FB, the most challenging classes in the dense model. This emphasizes the negative effects of regular pruning through IMP, which sacrifices weakly represented subgroups to maintain high overall accuracy. In contrast, SMS has a less significant impact on MB and FB but leads to a more nuanced decline in easier-to-classify subgroups.

Table 15: ResNet-18 on Celeb-A: Comparison of the pretrained (i.e. dense) base model against SMS and IMP for different sparsity levels, employing ALLR as the retraining schedule for 20 epochs of retraining. We indicate the top-1 test accuracy as well as the recall on the four different sub-groups. All results are averaged over multiple random seeds with standard deviation included.

**Celeb-A**

| Setting | Sparsity | Top-1 acc. | Balanced acc. | Sub-group Recall | | | |
|---|---|---|---|---|---|---|---|
| | | | | MB | MN | FB | FN |
| Pretrained | 0% | 98.99 ±0.01 | 98.09 ±0.02 | 94.41 ±0.20 | 99.93 ±0.00 | 98.48 ±0.00 | 99.54 ±0.00 |
| SMS | 90% | 99.02 ±0.00 | 98.13 ±0.04 | 94.52 ±0.20 | 99.92 ±0.00 | 98.57 ±0.02 | 99.50 ±0.04 |
| IMP | 90% | 98.99 ±0.02 | 98.12 ±0.03 | 94.52 ±0.20 | 99.93 ±0.01 | 98.51 ±0.04 | 99.52 ±0.03 |
| SMS | 95% | 98.91 ±0.01 | 98.08 ±0.05 | 94.74 ±0.20 | 99.88 ±0.01 | 98.53 ±0.07 | 99.17 ±0.06 |
| IMP | 95% | 98.79 ±0.01 | 97.89 ±0.08 | 94.02 ±0.31 | 99.92 ±0.01 | 98.20 ±0.01 | 99.41 ±0.01 |
| SMS | 97% | 97.79 ±0.08 | 96.04 ±0.53 | 89.44 ±1.78 | 99.67 ±0.07 | 97.38 ±0.24 | 97.68 ±0.49 |
| IMP | 97% | 97.74 ±0.08 | 95.73 ±0.32 | 87.42 ±1.17 | 99.88 ±0.01 | 96.71 ±0.07 | 98.90 ±0.04 |

## C  ABLATION STUDIES

### C.1  ABLATION: SMS HYPERPARAMETERS

We conduct several ablation studies to assess the influence of key hyperparameters in IMP: the retraining schedule and retraining length. Given the large number of individual runs in each ablation study, we restrict ourselves to examining WideResNet-20 trained on CIFAR-100. Besides the default pretraining parameters shown in Table 3, we employ a pretraining learning rate initiated at 1e-1, decaying by a factor of 0.2 at epochs 60, 120, and 160.

#### C.1.1  ABLATION: THE RETRAINING SCHEDULES

We begin by isolating the impact of the retraining schedule, comparing LRW, SLR, CLR, LLR and ALLR. Figure 9 depicts the difference between soup accuracy and best candidate accuracy for a wide range of sparsities in the One Shot setting, where we distinguish between $m = 3$ (left) and $m = 5$ (right). Note that each retraining schedule also influences the accuracy of candidate models. Throughout these experiments, the number of retraining epochs is fixed at 10.

First of all, we observe that all schedules except LRW effectively train pruned models to a state suitable for averaging. LRW, solely basing the initial learning rate magnitude on retraining duration, potentially falls short in recovering high pruning-induced performance degradation, thus hindering feasible averaging. Contrastingly, for all other schedules we see consistent improvements upon their averaging candidates, with strategies performing comparably well, although ALLR also augments performance in high sparsity scenarios. The lesser convergence of, for instance, SLR versus LLR, as identified by Zimmer et al. (2023), does not notably affect the accuracy disparity between soup and best model, even though LLR results in superior candidates and a better soup model.

We conclude that SMS requires retraining that is sufficiently accelerated by a proper learning rate schedule.

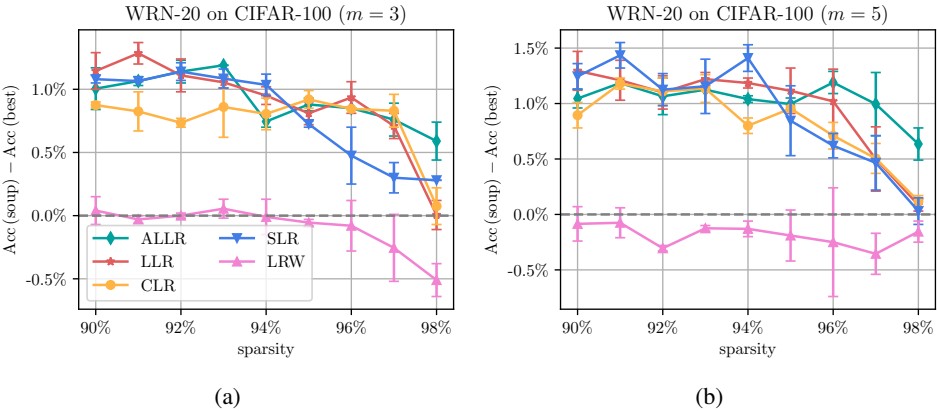

(a)                                                                 (b)

Figure 9: WideResNet-20 on CIFAR-100: Test accuracy difference between the soup of a) $m = 3$ or b) $m = 5$ models compared to the best candidate model for a wide range of sparsity levels. Each line depicts one retrain schedule. Note that we only consider the One Shot case and that the candidate models themselves depend on the retraining schedule at hand. Results are averaged over multiple random seeds with min-max bands indicated.

#### C.1.2  ABLATION: THE RETRAINING LENGTH

Next, we evaluate the impact of the retraining duration by comparing retraining lengths of 1, 2, 5, 10, and 20 epochs. As before, Figure 10 illustrates the accuracy difference between the soup and best candidate models across a spectrum of sparsity levels in the One Shot setting, differentiating between $m = 3$ (left) and $m = 5$ (right). We emphasize that the number of retraining epochs affects both the soup model accuracy as well as all candidate models for averaging. We stick to ALLR as the retraining schedule.

We encounter a diminishing returns scenario: the longer we retrain, the smaller the improvement of the soup upon the individual models. More surprisingly however, averaging models yields consistent improvements even with a mere single retraining epoch, which is clearly not enough for recovering performance in the high sparsity regime. The learning rate schedule ALLR seems to be of particular importance here, since it also incorporates the retraining length when choosing the learning rate schedule. As visible in Figure 9, such a consistent improvement is not achievable with other schedules, even when using 10 epochs of retraining.

We conclude that SMS is able to consistently improve upon the individual models even when using short amounts of retraining, provided that proper care is taken of the learning rate.

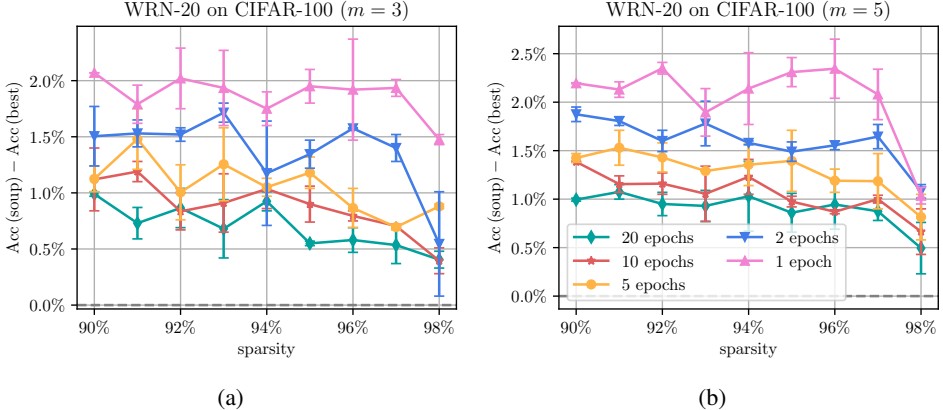

Figure 10: WideResNet-20 on CIFAR-100: Test accuracy difference between the soup of a) $m = 3$ or b) $m = 5$ models compared to the best candidate model for a wide range of sparsity levels. Each line depicts one retraining length configuration. Note that we only consider the One Shot case and that the candidate models themselves depend on the length of retraining at hand. Results are averaged over multiple random seeds with min-max bands indicated.

## C.2 ABLATION: SUITABLE BASELINES FOR SMS

We have demonstrated that SMS, which trains $m$ models per phase in parallel for $k$ epochs each, surpasses $\text{IMP}_{m\times}$ – a natural baseline where IMP retraining duration in each phase is extended by a factor of $m$ (totaling $k \cdot m$ epochs). In Table 16, we compare $\text{IMP}_{m\times}$ to another relevant baseline in the same setting as in Table 1: increasing the number of IMP phases by $m$, matching the total retraining epochs of SMS and $\text{IMP}_{m\times}$, but with a reduced pruning rate per phase. The results indicate that $\text{IMP}_{m\times}$, and consequently SMS, outperform this additional baseline.

Table 16: Comparison of test accuracy for different IMP baselines on ResNet-50 trained on ImageNet. Results are averaged over multiple seeds with standard deviation included.

| Method | $m = 3$ Accuracy | $m = 5$ Accuracy | $m = 10$ Accuracy |
|---|---|---|---|
| $\text{IMP}_{m\times}$ | 74.34% ±0.09% | 74.56% ±0.24% | 74.50% ±0.09% |
| IMP with $m$ phases | 73.69% ±0.10% | 74.08% ±0.04% | 74.70% ±0.02% |

## C.3 ABLATION: DIFFERENCES TO STOCHASTIC WEIGHT AVERAGING

Stochastic Weight Averaging (SWA, Izmailov et al. (2018)) is a popular procedure to improve the generalization performance of models by averaging their parameters along the training trajectory. In consequence, SWA and SMS are similar approaches, despite SWA being designed for dense models. We highlight some of the main observations and problems when combining SWA and IMP:

1. SWA is only beneficial if models of the same sparsity level and pattern are averaged, as differing sparsities will densify the model (see Figure 1). We hence apply SWA separately in each phase, starting each phase with the averaged model from the previous phase and reinitializing SWA accordingly.

2. In general, SWA and SMS are not excluding each other, they can be used in conjunction, potentially further improving the effect of SMS.

3. SWA requires either a cyclic or high constant learning rate to explore multiple optima for beneficial averaging. However, retraining after pruning uses specific translated learning rate schedules (such as FT, LRW, SLR, CLR, LLR or ALLR) to maximize performance.

Despite these issue of differing learning rate schedules, we conduct experiments using ResNet-50 on ImageNet, following the setup of Table 1 for a sparsity of 90% in three cycles. Precisely, Table 17 compares classical IMP to IMP with SWA, where we update the SWA-model after each epoch and set the retrained model to its averaged variant at the end of the phase as discussed above. We observe slightly inferior results when adding SWA, comparing a wide range of retraining learning rate schedules.

SWA is not able to improve the results of classical IMP (and hence also falls behind SMS by a large margin, cf. Table 1). We think that this is mostly due to the specific retraining schedules used for IMP, which stand in conflict with the requirements for SWA.

Table 17: ResNet-50 on ImageNet: Test Accuracy comparison of IMP (first row) vs. IMP with SWA (second row) for different retraining schedules when aiming for a goal sparsity of 90% in three cycles of ten retraining epochs each. Results are averaged over multiple seeds with standard deviation included.

| Method | FT | LRW | SLR | CLR | LLR | ALLR |
|---|---|---|---|---|---|---|
| IMP | 27.38% ±0.51% | 73.65% ±0.08% | 73.29% ±0.07% | 73.36% ±0.02% | 73.38% ±0.25% | 73.80% ±0.10% |
| IMP + SWA | 22.11% ±0.36% | 73.34% ±0.08% | 72.10% ±0.02% | 72.19% ±0.18% | 72.25% ±0.11% | 73.01% ±0.04% |

## C.4 ABLATION: PERFORMANCE DEGRADATION FOR EXTREME LEVELS OF SPARSITY

We have argued that extremely high sparsity levels lead to a model that is not stable to randomness anymore, i.e., two retrained models do not lie in the same basin and thus cannot be averaged. For WRN-20 on CIFAR-100, this problem occurs at 99% pruned in One Shot and above, see Figure 4a where the uniform approach is unable to average the models with increasing performance.

To investigate this issue, we track the $L_2$-norm distance between the candidates for averaging. Table 18 displays the mean and maximal $L_2$ distance between each pair of five candidates for averaging, using One Shot pruning and retraining in the same setting as in Figure 4a. We observe that for sparsities in the range 90%-98%, the mean and maximal $L_2$-distance between the five candidate models are relatively stable among sparsities. Increasing the sparsity to 99% and 99.5% however leads to a much increased distance between the retrained models. At this sparsity, the models are driven further apart, supporting our hypothesis of instability to randomness - they do not converge to the same basin.

Table 18: Mean (first row) and maximal (second row) $L_2$-distance when comparing each pair of the five candidates for averaging in Figure 4a for different sparsity levels between 90% and 99.5%. Results are averaged over multiple seeds, where we omit the standard deviation for the sake of clarity.

| Sparsity | 90% | 92% | 94% | 96% | 98% | 99% | 99.5% |
|---|---|---|---|---|---|---|---|
| mean $L_2$-distance | 29.17 | 29.44 | 29.61 | 29.66 | 30.13 | 32.95 | 39.41 |
| max $L_2$-distance | 29.22 | 29.48 | 29.69 | 29.70 | 30.24 | 33.35 | 40.18 |

