# OpenReview forum: "Sparse Model Soups: A Recipe for Improved Pruning via Model Averaging"
_ICLR.cc/2024/Conference — ICLR 2024 poster_

### Official Review · Reviewer_hicK · 2023-10-28

**Soundness:** 3 good
**Presentation:** 3 good
**Contribution:** 3 good
**Rating:** 6
**Confidence:** 4

**Summary:**

This paper introduces a new pruning technique named Sparse Model Soups, which combines the weight averaging methods from [1] with the pruning method described in [2]. They provide empirical evidence to support the idea that this straightforward aggregation enhances the performance of pruned models in image classification tasks.

References

[1] Mitchell Wortsman, Gabriel Ilharco, Samir Ya Gadre, Rebecca Roelofs, Raphael Gontijo-Lopes, Ari S Morcos, Hongseok Namkoong, Ali Farhadi, Yair Carmon, Simon Kornblith, et al. Model soups: averaging weights of multiple fine-tuned models improves accuracy without increasing inference time. In International Conference on Machine Learning, 2022.

[2] Song Han, Jeff Pool, John Tran, and William Dally. Learning both weights and connections for efficient neural networks. In C. Cortes, N. Lawrence, D. Lee, M. Sugiyama, and R. Garnett (eds.), Advances in Neural Information Processing Systems, 2015.

**Strengths:**

Clarity
- The paper is well written, ensuring it is understandable for readers.
- The suggested method is straightforward and effectively explained for easy understanding.

Originality and Significance
- This paper introduces a novel pruning method that incorporates model averaging techniques based on the Model Soup methods.
- They provide empirical evidence demonstrating that the proposed method offers improved performance when compared to baseline approaches.

**Weaknesses:**

Method
- Although they can parallelize the training process, performing $m\times k$ training epochs still imposes a substantial computational burden. And if the training cost becomes small, the overall performance gain significantly drops for the extreme sparsity cases.

Experiments
- It would be valuable to conduct empirical analyses to investigate why performance degradation occurs in regions of extreme sparsity.
- Similarly, it would be beneficial to empirically analyze why SMS performs well in situations of early sparsity, where batch randomness can lead to divergence between averaged weights [3].
- It would be advantageous to include an ablation study exploring different combinations of averaging coefficients where the $\lambda_i$ values differ from each other.

Recommend
- I suggest that the authors consider adding an Ethics Statement and a Reproducibility Statement immediately following the main paper.

References

[3] Behnam Neyshabur, Hanie Sedghi, and Chiyuan Zhang. What is being transferred in transfer learning? Advances in neural information processing systems, 2020.

**Questions:**

See the weakness section.

---

> ### Author Response · Authors · 2023-11-15
> **Response to Reviewer hicK**
>
> We thank you for your positive feedback and constructive remarks.
>
> ### Weaknesses
> > Although they can parallelize the training process, performing $m \times k$ training epochs still imposes a
> > substantial
> > computational burden. And if the training cost becomes small, the overall performance gain significantly drops for
> > the extreme sparsity cases.
>
> We acknowledge that all model-averaging methods that rely on training multiple models in parallel (as e.g. the
> original model soups) induce some computational overhead in order to increase the final performance. Approaches that
> average models along the training trajectory (such as e.g. SWA) do not have that drawback, but fall behind in
> performance, at least in our particular setting (please see the response to Reviewer fESa for a full account on
> methods like SWA).
>
> However, please note that this is only true for extreme levels of sparsity, that would require iterative pruning in
> any case to reach the full performance of classical IMP. In fact, as visible in Figure 4b, for reasonable levels of
> sparsity SMS is able to outperform its prolonged IMP counterpart after a minimum number of epochs and with
> significant improvements. In Appendix C, we further study that SMS is able to improve among its individual averaging
> candidates independent of the retraining length and sparsity - there just exists a certain tradeoff at which
> retraining is too short such that averaging $m$ models gives better results than retraining $m$ times as long.
>
> > It would be valuable to conduct empirical analyses to investigate why performance degradation occurs in regions of
> > extreme sparsity.
>
> Thank you for that remark - we have addressed this point from a qualitative perspective, but we agree that further
> empirical substantiation improves our work. In the paper, we argue that extremely high sparsity levels (the problem
> occurs at 99\% pruned in One Shot and above) lead to a model that is not stable to randomness anymore, i.e., two
> retrained models do not lie in the same basin and thus cannot be averaged. To substantiate this claim, we have run
> further experiments and tracked several metrics to see how the candidates for averaging behave. One important
> empirical observation lies in tracking the $L_2$-norm distance between the candidates: we observe that for
> sparsities in the range 90\%-98\%, the mean and maximal $L_2$-distance between the 5 candidate models as in the
> experiment for Figure 4a are relatively stable among sparsities. Increasing the sparsity to 99\% and 99.5% however
> leads to a much increased distance between the retrained model. At this sparsity, the models are driven further
> apart, supporting our hypothesis of instability to randomness - they do not converge to the same basin.
>
> The upcoming revision will include an ablation study including these experiments in a more detailed form. We will
> include them in the manuscript as soon as possible and will let you know.
>
> > Similarly, it would be beneficial to empirically analyze why SMS performs well in situations of early sparsity,
> > where batch randomness can lead to divergence between averaged weights [3].
>
> Could you elaborate what you mean by 'early sparsity'? We are not entirely sure if we understand you correctly here.
>
> > It would be advantageous to include an ablation study exploring different combinations of averaging coefficients
> > where the $\lambda_i$ values differ from each other.
>
> We have experimented a lot with what Wortsman et al. call *LearnedSoup*, where the $\lambda_i$ coefficients are
> learned through optimization on a held-out validation set. In our experiments, LearnedSoup was never able to reach
> the performance of UniformSoup or GreedySoup. We are happy to include these results in the appendix in the next
> revision - do you have any particular suggestion on how to perform an ablation study on the effect of differing
> $\lambda_i$? How should the values be chosen in such a study, if not automatically as in LearnedSoup?
>
> ### Recommendations
> > I suggest that the authors consider adding an Ethics Statement and a Reproducibility Statement immediately
> > following the main paper.
>
> Thanks for that remark, we will add both to the revision.
>
> We thank you again for your insightful review and hope to have addressed your concerns. Please let us know in case
> there are further doubts or questions.

---

> > ### Comment · Reviewer_hicK · 2023-11-16
> > **Additional question**
> >
> > Thank the authors for the detailed explanation.
> >
> > Similarly, it would be beneficial to empirically analyze why SMS performs well in situations of early sparsity, where batch randomness can lead to divergence between averaged weights [3].
> >
> > Here, what I mean by 'early sparsity' is when a lower sparsity level in IMP iterations occurs (which are early iteration stages in IMP). In such cases, batch randomness can cause divergence among solutions, creating a high loss barrier between particles. This divergence could impede the particles from being effectively averaged.
> >
> > And, I am curious how the SMS can perform well in that situation.

---

> > > ### Author Response · Authors · 2023-11-16
> > > **Reponse to additional question**
> > >
> > > Thanks for the quick response and clarification!
> > >
> > > Our experiments align with our intuition that the case of low pruning ratio (as in e.g. what you call early stages) is not nearly as detrimental to the overall performance of SMS as the high or extreme pruning ratios. In fact, SMS deals perfectly with low sparsities, the issue of batch randomness causing divergence (what we call 'instability to randomness') is more of a problem in the high sparsity regime. Please see for example Figure 4a: in the low/medium sparsity regime (70\%), SMS yields an immediate over the individual models and with increasing sparsity, this benefits starts to decrease. Further, see Figure 4b, where the green line corresponds to a sparsity of 90\% (which is high, but not extreme in that setting): Even with as little as retraining for a single epoch, the soup is better than performing $IMP_{m\times}$. We refer also to the ablation study in Appendix C, showing that these observations similarly hold for various retraining schedules and lengths.
> > >
> > > To sum up: Instability to batch randomness is more of an issue for high sparsities. That is also intuitive and it is not that clear how to empirically analyze why SMS performs well in that scenario. Our argument in the paper is more of a qualitative than quantitative nature: Frankle et al. have shown that there is a point at which sufficient pretraining has been done such that the model can be split into two individually trained branches, which end up in the same loss basin and are thus averageable. That means, starting from a randomly initialized model is not enough to obtain stability, but after some amount of pretraining, the network is stable. In our case, the network has been fully pretrained to convergence. Suppose we would not prune (i.e. prune to a sparsity of of 0\%), then with the results of Frankle et al. in mind, it is clear that the retrained models are averageable (assuming a reasonable learning rate schedule). With increasing sparsity, the model gets more and more damaged and at some point the individual branches start to diverge. However, in the low sparsity regime, the pruned model is still very stable and batch randomness is not able to let the retrained models diverge.
> > >
> > > Does that answer your question, or is further clarification needed? Please let us know and we are happy to clarify further. Thanks!

---

> > > > ### Comment · Reviewer_hicK · 2023-11-18
> > > > **Response to Authors**
> > > >
> > > > Your extra explanations are appreciated. I intend to maintain my score comfortably above the acceptance threshold.

---

### Official Review · Reviewer_fESa · 2023-11-01

**Soundness:** 2 fair
**Presentation:** 3 good
**Contribution:** 2 fair
**Rating:** 6
**Confidence:** 2

**Summary:**

This paper presents the Sparse Model Soups (SMS) framework, which applies the model soup algorithm to the neural network pruning procedure. Experimental results show that the model soup algorithm, a one of the most well-known weight-averaging methodologies that had demonstrated notable success in training dense neural networks, is also effective in iterative pruning procedures for training sparse neural networks.

**Strengths:**

Originality and significance: The proposed SMS framework does not bring a substantial level of novelty, as one can consider it as an application of the existing model soup algorithm to an arbitrary iterative pruning process. However, its originality comes from the actual implementation in the domain of neural network pruning, even though the individual elements may already exist separately. Considering the recent success of weight averaging methods for dense neural networks, it is valuable to explore the extension of these techniques to sparse neural networks.

Quality and clarity: Based on observations in the field of transfer learning, where fine-tuned models from the same pre-trained model tend to stay in the nearby region, the hypothesis that a similar phenomenon will occur when re-training the same pruned model is well-founded. The effectiveness of the proposed SMS framework is confirmed through a range of experiments conducted in the domains of image classification, semantic segmentation, and neural machine translation.

**Weaknesses:**

While the proposed SMS framework incorporates the model soup algorithm in the context of neural network pruning, it does not provide specific insights into the unique factors that are especially pertinent to sparse network training. The questions remain: What attributes of the model soup algorithm contribute to its effectiveness in the neural network pruning regime? Is it the same reason why weight-averaging methods have succeeded in conventional dense network training?

**Questions:**

1. Alongside model soups, another prominent weight-averaging strategy is Stochastic Weight Averaging (SWA). The fact that SMS requires m training runs at each cycle makes SWA, which performs weight averaging within a single SGD trajectory, somewhat appealing. Are there any baseline results using SWA instead of model soups?

2. I understand that the authors have opted to show exclusively the UniformSoup results for CityScapes and WMT16 since the GreedySoup algorithm utilizes validation data, which are the test split here. However, despite the potential fair comparison issue, presenting additional GreedySoup results might offer valuable insights into the benefits of selectively using soup ingredients at high sparsity levels.

---

> ### Author Response · Authors · 2023-11-15
> **Response to Reviewer fESa**
>
> Thank you for your review! We are happy to address the individual points raised, as they help us to improve our work.
>
> ### Weaknesses
> > What attributes of the model soup algorithm contribute to its effectiveness in the
> > neural network pruning regime? Is it the same reason why weight-averaging methods have succeeded in conventional
> > dense network training?
>
> To clarify, let us re-emphasize the key insight of our work. First of all, addressing your second question,
> weight-averaging methods that "have succeeded in conventional dense network training" (such as e.g. SWA) are of a
> different nature: they average models along the optimization path to explore different optima but always train a
> single model with fixed hyperparameters.
>
> In contrast, the less conventional model soup approach is specific to transfer learning and involves training
> and averaging multiple models with varied hyperparameters. Its effectiveness in transfer learning has been
> thoroughly validated by Wortsman et al., with their empirical evidence (Figure 2 in Wortsman et al., 2022a) showing
> the best generalizing solution often resides 'between' two finetuned networks.
>
> Returning to our work, we maintain the same downstream task without domain shifts. However, we think and have argued
> that pruning and retraining a model strongly resembles the finetuning paradigm in transfer learning: we interpret
> pruning as feature perturbation and retraining as adapting these features to the original task, as if the original
> pretraining would have been done on a different task (i.e. the one leading to the perturbed features). While we
> think that this provides a good intuition, it is far from obvious that the model soup mechanism also holds when a
> large portion of weights are pruned, reducing the model's performance to that of a random classifier. The model
> undergoes significant
> perturbation/damage, raising questions about its 'stability to randomness' (as per Frankle et al. 2020). It's
> uncertain if two versions of such a heavily pruned model would be averageable, particularly as aggressive learning
> rate schedules during retraining could drive the models apart.
>
> Our work is the first to empirically demonstrate that this is indeed possible and significant performance gains,
> both in-distribution and out-of-distribution, are achievable across diverse architectures and datasets. Returning to
> your first question, we think that the insights from transfer learning can be transferred to
> the network pruning regime, motivated by the resemblance between the two paradigms as outlined above, and substantiated
> by extensive results.
>
> ### Questions
> > Are there any baseline results using SWA instead of model soups?
>
> Thank you for suggesting the comparison between SWA and SMS, it's an interesting improvement to our work. We
> conducted experiments, but before reporting and discussing the results, let us make some general remarks.
>
> 1. SWA is only beneficial if models of the same sparsity level and pattern are averaged, as differing sparsities
>    will densify the model (see Figure 1). We apply SWA separately in each phase, starting each phase with the averaged model from
>    the previous phase and reinitializing SWA accordingly.
> 2. SWA and SMS are not excluding each other, they can be used in conjunction, potentially further improving the
>    effect of SMS.
> 3. SWA requires either a cyclic or high constant learning rate to explore multiple optima for beneficial averaging.
>    However, retraining after
>    pruning uses specific translated learning rate schedules (such as FT, LRW, CLR, LLR or ALLR) to maximize
>    performance.
>
> In our experiments using ResNet-50 on ImageNet, following the same setup as in Table 1 for a sparsity of 90\% in
> three cycles,we observed slightly inferior results with SWA:
>
> - IMP: Test Accuracy 73.72\% ($\pm$ 0.00\%)
> - IMP+SWA: Test Accuracy 73.05\% ($\pm$ 0.02\%)
>
> SWA is not able to improve the results of classical IMP (and hence also falls behind SMS by a large margin, cf.
> Table 1). We think that this is mostly due to the specific retraining schedules used for IMP, which stand in
> conflict with the requirements for SWA. We will add an ablation study dissecting these factors as soon as possible.
>
>
> > [...] despite the potential fair comparison issue, presenting additional GreedySoup results might offer valuable
> > insights into the benefits of selectively using soup ingredients at high sparsity levels.
>
> We are happy to create a separate validation split for these two specific tasks and will add the experiments as
> soon as possible, however we do not think that GreedySoup will be able to offer much of an improvement - apart from
> extreme levels of sparsity, the uniform approach lead to the most consistent improvements.
>
> Thanks for your review, we hope to have addressed your concerns and answered your questions. Please let us know if
> further clarification is needed.

---

> > ### Comment · Reviewer_fESa · 2023-11-22
> >
> > Thank you for the authors' efforts in addressing my concerns.
> >
> > 1. Regarding the SWA baseline: The observed performance decline of IMP + SWA compared to IMP is likely due to insufficient hyperparameter tuning, as per my experimental experience. I acknowledge the time limitations during the rebuttal, making comprehensive experiments challenging. I expect that future revisions will provide more thorough experiments and insightful discussions. I am inclined to place the trust in the authors' commitment, as stated in their response: "We will add an ablation study dissecting these factors as soon as possible."
> >
> > 2. Regarding the GreedySoup on CityScapes and WMT16: I thought the experiment would be simple, applying the GreedySoup algorithm to the ingredients already employed for the UniformSoup results provided by the authors. Although the authors said, "we will add the experiments as soon as possible," the end of the discussion period is approaching. I am curious if there are any issues or complications with the experiment.

---

> > > ### Author Response · Authors · 2023-11-22
> > >
> > > Thank you for taking the time to respond to our rebuttal!
> > >
> > > > Regarding the SWA baseline:
> > >
> > > We agree that this performance decline might be due to suboptimally chosen hyperparameters, especially the learning rate schedule during retraining. As noted in our initial rebuttal, we think that tuning the learning rate for SWA might itself be in conflict with the specific dynamics of retraining after pruning. We are happy to investigate this, however we think that doing a comprehensive study on this setting will not be possible within the discussion period -  we plan to include it in a potential camera ready version. In any case, assuming that we are able to tune IMP+SWA to yield improvements over IMP, we see no reason why also adding SWA to SMS should yield worse results. We expect SWA to be beneficial to SMS. Further, we do not expect SWA to reach close to the same results as standalone SMS. For non-sparse model soups, this was also investigated thoroughly by Wortsman et al., we are confident that these results transfer.
> > >
> > > > Regarding the GreedySoup on CityScapes and WMT16:
> > >
> > > There were indeed some complications with respect to the CityScapes experiments, as we outline below. To deliver the results for GreedySoup, we have to define an additional validation set. For WMT16, we were able to utilize a validation dataset available through Huggingface, allowing us to simply use the previously computed ingredients. For CityScapes, there is no additional validation set freely available, we have to split either from the train or test set - in any case this changes all previous results, requiring recomputation of all CityScapes experiments in order to provide consistent results. Since these experiments are still running, we have not yet updated the revision and will do so as soon as the CityScapes results are available.
> > >
> > > However, to address your question, we included the result for WMT-16 below. GreedySoup is not able to give an improvement over the uniform approach, as we expected. In previous experiments, we noticed that it is almost never beneficial to greedily select when the random seed is varied. This is definitely not the case when varying other hyperparameters: say e.g. we vary the weight decay and pick a too large weight decay value for some of the runs - in that particular case, the GreedySoup may improve upon the UniformSoup approach.
> > >
> > > | **WMT-16** | **Sparsity 50.0\% (One Shot)** | **Sparsity 60.0\% (One Shot)** | **Sparsity 70.0\% (One Shot)** |
> > > |---|---|---|---|
> > > | **Accuracy of** | $m=3$ | $m=3$ | $m=3$ |
> > > | SMS (uniform) | 25.47 ± 0.52 | **25.09 ± 0.00** | **24.51 ± 0.43** |
> > > | best candidate | 25.39 ± 0.03 | 24.96 ± 0.26 | 24.12 ± 0.01 |
> > > | mean candidate | 25.16 ± 0.08 | 24.79 ± 0.19 | 24.03 ± 0.01 |
> > > | SMS (greedy) | **25.51 ± 0.28** | 24.92 ± 0.47 | 24.14 ± 0.02 |
> > > | best candidate | 25.39 ± 0.03 | 24.96 ± 0.26 | 24.12 ± 0.01 |
> > > | mean candidate | 25.16 ± 0.08 | 24.79 ± 0.19 | 24.03 ± 0.01 |
> > > | $IMP_{m\times}$ | 25.36 ± 0.12 | 25.09 ± 0.05 | 24.00 ± 0.04 |
> > > | IMP | 25.15 ± 0.20 | 24.90 ± 0.20 | 24.04 ± 0.28 |
> > >
> > > We hope that this resolves the issue - we will update the revision as soon as we have consistent and clear results for CityScapes. At this point, it makes no sense to simply apply GreedySoup to the previously computed ingredients, since by either splitting a validation set from train or test (i.e. the actual validation dataset we evaluate on), we have to adjust the results for all other IMP variants and the UniformSoup setting as well.

---

> > > > ### Comment · Reviewer_fESa · 2023-11-22
> > > >
> > > > I am thankful for the prototype results that were presented to address my concerns, and I expect dealing with them more extensively in subsequent revisions. Consequently, I am increasing the score to six.

---

### Official Review · Reviewer_ja8N · 2023-11-02

**Soundness:** 2 fair
**Presentation:** 3 good
**Contribution:** 2 fair
**Rating:** 6
**Confidence:** 4

**Summary:**

The idea is to combine model soups from transfer learning with pruning. The proposal is : in prune-retrain paradigm (be it from scratch or pretrained), the training portion is replaced by model-soup. It improves the overall quality of the final pruned model as is validated extensively in experiments.

[Update]
In light of the responses by authors, I am increasing my score to 6.

**Strengths:**

This being an empirical paper with simple idea proposal, the authors do an excellent job of evaluating their idea in terms of
(1) extensive experiments on different domains
(2) covering baselines that are natural competitors to the proposal.

**Weaknesses:**

1) I am not sure if this paper contributes new ideas or analysis. The proposal is to replace the training portion of prune-retrain with model soups. I do not have background on transfer learning, but as a general machine learning person, it is not surprising that it improves the accuracy of the model given the backdrop of model soups paper . Since in both the cases it holds that m copies of model start from the same initialization.

**Questions:**

1) How is IMP $m \times$ implemented? Is the pruning rate for each prune step reduced? or is the training portions increased m$\times$. The latter, I suspect, will not be very useful.
2) Are there any challenges that are specific to using model soups for training portion of prune-retrain algorithm which differentiate it from applying model soups to finetuning of pretrained models? I felt that there were no new challenges here.

---

> ### Author Response · Authors · 2023-11-15
> **Response to Reviewer ja8N**
>
> We thank you for reviewing our work and acknowledging its strengths! Let us respond to your concerns and questions
> in detail.
>
> ### Weaknesses
> > I do not have background on transfer learning, but as a general machine learning
> > person, it is not surprising that it improves the accuracy of the model given the backdrop of model soups paper .
> > Since in both the cases it holds that m copies of model start from the same initialization.
>
> We regret to hear your doubts about the novelty of our work and wish to clarify. To begin with, the averageability of
> models trained from identical initializations is not obvious. Neyshabur et al. (2020) observed that models with the
> same random
> initialization but varying batch selections diverge and are not averageable, whereas Frankle et al. (2020)
> established averageability after starting from a sufficiently pretrained model. Wortsman et al. (2022a) found that
> finetuning models from a single pretrained model are averageable,
> despite task changes (i.e. transfer learning) and the pretrained model potentially being bad at the new task.
>
> In our research, we avoid domain shifts, focusing on the same downstream task. In non-pruning scenarios, it's
> somewhat clear that extended training with a reasonable learning rate might yield models whose average is not worse than individual models. However, this is far from obvious in heavily pruned models, where performance drops
> significantly. The substantial damage these models undergo raises questions about their 'stability to randomness'
> (as per Frankle et al. 2020). It's not evident that two heavily pruned models are averageable, especially
> under aggressive retraining schedules.
>
>
> Our work is the first to empirically show that averaging such models leads to notable performance improvements in-
> and out-of-distribution across various architectures and datasets. We link pruning and retraining to finetuning in
> transfer learning, treating pruning as feature perturbation and retraining as features adaption to the original task
> (as if pretraining would have lead to the perturbed features). A key challenge was overcoming the decreased sparsity
> from naive weight averaging due to different sparsity patterns (cf. Figure 1), requiring retraining at the expense of
> performance (cf. IMP-RePrune, Table 1). Our
> solution, SMS, effectively addresses this issue. Therefore, we respectfully disagree that our papers does
> not contribute new ideas, as outlined.
>
>
> ### Questions
> > How is $IMP_{m\times}$ implemented? Is the pruning rate for each prune step reduced? or is the training portions
> > increased m times. The latter, I suspect, will not be very useful.
>
> $IMP_{m\times}$ increases training epochs per prune-retrain-cycle by a factor of $m$. We believe this, arguably most
> natural baseline is useful; could you elaborate on why you think it isn't?. Nevertheless, we
> agree that reducing the pruning rate accordingly (i.e. for $m=3$, triple the
> amount of prune-retrain cycles to reach a final sparsity) is an interesting
> baseline, thank you for that remark! In our experience, lower
> pruning rates do not necessarily enhance IMP's final product; in fact, empirical evidence from Bartoldson et al.
> indicates that the instability from significant pruning can actually aid generalization.
>
> In response to your question, we conducted experiments with ResNet-50 on ImageNet, aiming for a 90% sparsity goal,
> replicating the setting in Table 1. We compared the final test accuracies of $IMP_{m\times}$, which extends retraining
> length per cycle, with a variant increasing the number of cycles by $m$ but maintaining a 10-epoch retraining length
> per cycle. The results are as follows:
>
> | Method | $m=3$ | $m=5$ | $m=10$                 |
> |---|---|---|------------------------|
> | $IMP_{m\times}$ | 74.34\% ($\pm$ 0.09\%) | 74.56\% ($\pm$ 0.24\%) | 74.50\% ($\pm$ 0.09\%) |
> | IMP with $m$ phases | 73.69\% ($\pm$ 0.10\%) | 74.08\% ($\pm$ 0.04\%) | 74.70\% ($\pm$ 0.02\%) |
>
> This approach mostly falls behind $IMP_{m\times}$ and in consequence also behind SMS. We however thank you for
> suggesting this additional baseline, we will include it in the manuscript.
>
> > Are there any challenges that are specific to using model soups for training portion of prune-retrain algorithm
> > which differentiate it from applying model soups to finetuning of pretrained models?
>
> We hope that this concern is sufficiently addressed by our explanations regarding the novelty and contribution of
> our work - the setting is different and it is a priori not clear whether aggressive pruning yields
> models that are averageable, especially given the existing (aggressive) learning rate schemes for retraining.
> Further, it is a priori not clear how to mitigate the problem of reduced sparsity when averaging arbitrary sparse
> models; a problem that does not exist in the setting of transfer learning.
>
> Thanks again for your review - please let us know if further clarification is needed!

---

### Official Review · Reviewer_TvoY · 2023-11-05

**Soundness:** 1 poor
**Presentation:** 3 good
**Contribution:** 1 poor
**Rating:** 5
**Confidence:** 4

**Summary:**

The paper proposes merging sparse models by initiating each prune-retrain cycle with the averaged model from the previous phase. They show that averaging these models significantly enhances generalization and OOD performance over their individual counterparts. Overall, in summary, it is an extension of model soups for sparse models.

**Strengths:**

1. The experimental section of the paper + supplementary is rich illustrating noticeable gain.
2. OOD experiments are new for sparse model soup showing SMS consistently improves over the baselines.

**Weaknesses:**

I have significant novelty concerns with the draft.

1. The idea of sparse model averaging has been widely explored including the model soups (eg. https://arxiv.org/abs/2205.15322 https://arxiv.org/abs/2208.10842 https://arxiv.org/abs/2306.10460  etc).
2. The authors have failed to detail how their method contrasts with existing sparse model soup papers in their related work section. I feel it is just an incremental work over the existing literature. The benefits of averaging the sparse masks is already known.
3. Although I appreciate the extensive experiments by authors, I still feel the experiments are limited to small-scale datasets and models (maybe ViT scale or OPT models-based experiments will add value).
4. I feel auxiliary benefits of model soups like OOD robustness, fairness, etc are good directions to explore.

**Questions:**

See above.

---

> ### Author Response · Authors · 2023-11-15
> **Response to Reviewer TvoY**
>
> Thank you for reviewing our manuscript. In the following, let us address your concerns in detail.
>
> ### Weaknesses
>
> > 1. The idea of sparse model averaging has been widely explored including the model soups (eg. https://arxiv.
> > org/abs/2205.15322 https://arxiv.org/abs/2208.10842 https://arxiv.org/abs/2306.10460 etc).
>
> We are surprised to see the novelty of our work being questioned, especially since we cite all the papers you
> mention. We regret if this misunderstanding stems from poor communication from our side, see also the
> response to your second concern below. Let us clarify how our work differs from the cited papers:
>
> **2205.15322: Yin et al.** focus on Dynamic Sparse Training (DST), which is different from our domain (prune-after-
> and -during-training). They average models within a single training run using fixed hyperparameters, not across
> multiple runs. Their averaging destroys the sparsity (cf. Figure 1), requiring re-pruning: this is inherent to their
> prune-and-grow DST approach, requiring to explore different sparsity patterns. To mitigate the negative impacts of
> re-pruning, they propose averaging methods like *CIA* and *CAA*. In contrast, our work deliberately
> avoids re-pruning by keeping consistent sparsity patterns. Note that **we explicitly compare to the re-pruning
> approach** (cf. Table 1, IMP-RePrune), using both their methods CIA and CAA, outperforming their re-pruning schemes
> significantly.
>
> **2208.10842: Yin et al.** use IMP with weight rewinding (IMP-WR) and concentrate on generating lottery tickets, a
> distinct goal from ours of creating high-performing sparse models. Weight rewinding rewinds the networks to the
> point of stability (as per Frankle et al.), the issue whether adjacent models are
> averageable is thus entirely
> different to whether they are averageable without rewinding. While the authors average models with varying
> sparsity (those adjacent in the lottery ticket generation process) necessitating re-pruning, our method averages
> parallely trained models, avoiding this detrimental step in our setting.
>
> **2306.10460: Jaiswal et al.** do not average model parameters at all, but rather average early pruning
> masks to quickly generate subsequent masks. This approach doesn't overlap with our contributions, focusing on
> fast mask generation rather than averaging sparse models.
>
> If there are other publications you believe align closely with our work, especially those implied by "etc." in your
> list, could you please specify them?
>
> > 2. I feel it is just an incremental work over the existing literature. The benefits of
> > averaging the sparse masks is already known.
>
> We were unaware that our manuscript could be misunderstood in that regard, but we take your concern seriously and
> will clarify the differences with the mentioned papers in our revised manuscript, which will be available soon. We
> hope that this resolves the issue.
>
> We respectfully disagree that our work is incremental. To our knowledge, no other publication
> has explored sparse model averaging in the context of prune-after-training or
> during-training; in fact, "averaging the sparse masks" is entirely different from what we do (cf. our
> explanations regarding Jaiswal et al.). Secondly, previous works on averaging sparse models involve a single
> training run and fixed hyperparameters, unlike our model soup approach that averages models from multiple runs with
> varying hyperparameters. It has been far from
> obvious that this was possible given that pruning drastically alters the model and reduces stability to randomness.
> Finally, our approach uniquely preserves the sparsity pattern, avoiding the need for repruning, a step we
> demonstrate as harmful and which we significantly improve upon.
>
> > 3. Although I appreciate the extensive experiments by authors, I still feel the experiments are limited to
> > small-scale datasets and models (maybe ViT scale or OPT models-based experiments will add value).
>
> We respectfully note that our experiments extend beyond "small-scale datasets and models." In fact, they encompass
> models of "ViT scale", including extensive experimentation with the vision-transformer MaxViT on ImageNet and the
> T5-transformer on WMT16, detailed in Appendix B.
>
> > 4. I feel auxiliary benefits of model soups like OOD robustness, fairness, etc are good directions to explore.
>
> We have conducted thorough experiments regarding the benefits of SMS in the OOD (Appendix B.2.3) and fairness
> (Appendix B.2.4) setting, demonstrating a consistent and significant improvement over the baseline, particularly in the
> OOD-case. Could you elaborate what additional directions you think are worth exploring?
>
> We hope our responses have clarified the contributions and novelty of our work. If you
> have any more questions or concerns, or suggestions for improvement, please let us know. Otherwise, we kindly ask
> you to reconsider your initial evaluation.

---

> > ### Comment · Reviewer_TvoY · 2023-11-22
> > **Post rebuttal response**
> >
> > After reviewing the authors' responses, I can find some distinguishing factors between the past literature and this paper's method. I again went through the new experiments added in the rebuttal.  I raise my score to 5 based on the author's rebuttal despite having some reservations about novelty.

---

### Author Response · Authors · 2023-11-20
**Official comment regarding the revision**

We thank all reviewers for their valuable feedback. We have revised our manuscript, incorporating a new ablation study on an additional baseline that increases the number of phases in IMP by a factor of $m$ (see Appendix C, addressing Reviewer ja8N's comments). Additionally, we have enhanced the related work section, offering a clearer distinction of our work from existing sparse averaging research, in response to Reviewer TvoY.

In our individual responses, we have conducted further experiments to address your concerns and are prepared to incorporate these findings into the manuscript. We believe these revisions adequately address the points raised and invite your feedback on the rebuttal. Your input is especially crucial as we approach the end of the discussion period. We look forward to your thoughts and suggestions for any additional clarifications. Thank you again!

---

### Author Response · Authors · 2023-11-23
**Official Comment regarding the end of the Discussion Period**

As the discussion period ends, we again would like to express our gratitude to all reviewers for their constructive feedback, assistance in improving our work and for increasing their scores accordingly.

For the sake of clarity, let us quickly summarize the findings of the discussion period and changes in the latest revision, as well as the changes that will be included in the final revision.

**Current revision:**
- Inclusion of an ablation study examining an additional baseline variant of IMP.
- An updated related work section, offering clearer distinctions from existing methodologies.

**Updates for the final revision:**
- A detailed ablation study on the effects of SWA on SMS.
- Incorporation of GreedySoup experiments for the CityScapes and WMT datasets. WMT results were shared in response to Reviewer fESa, while CityScapes experiments for the greedy approach are currently being done.
- An additional ablation providing more empirical evidence on the divergence of models at extremely high sparsity levels, as preliminarily discussed in response to Reviewer hicK.
- The reproducibility and ethics statement.

---

### Meta-Review · Area_Chair_ohqF · 2023-11-27

**Metareview:**

The reviewers and meta reviewer all carefully checked and discussed the rebuttal. They thank the authors for their response and their efforts during the rebuttal phase.

The reviewers and meta reviewer all acknowledge that the submission studies a topic relevant to the ICLR community, namely constructing sparsity-preserving parameter-based ensembles.
The response helped clarify several concerns, e.g., (i) novelty and positioning with respect to the previous literature, (ii) significance in the sense of whether the application of model soup was trivially expected to work, and (iii) the scale of the experiments. Some further changes have been promised during the rebuttal.

As a result, the reviewers and the meta reviewer are weakly inclined to accept the paper.
In particular, the author(s) are urged to carefully update their final manuscript with the following points:

* Tighten up the discussion with respect to previous works
* Better explain why model soup is not expected to work out of the box in this setting
* Integrate the new (some of which, still pending) results about SWA, uniform vs. greedy selection,  and another baseline (IMT with m steps)
* Add the analysis about the divergence in the extreme sparsity-level regime

If the paper was submitted to a journal, it would be accepted conditioned on those key changes, the meta reviewer thus expects all those changes to be carefully implemented.

**Justification For Why Not Higher Score:**

* The scope of the paper may not impact the greater ICLR community
* The innovation of the paper may be limited in the sense that it is more about using different existing building blocks (IMP, model soup) and carefully analyzing the combined effect.
* More insights would be beneficial to better understand the deeper reasons why model soup continues to work in that sparse regime

**Justification For Why Not Lower Score:**

* Well presented and overall clear manuscript
* Good empirical results and rigorous ablation studies/evaluation protocols

---

### Decision · Program_Chairs · 2024-01-16

Accept (poster)